# The Score-Difference Flow for Implicit Generative Modeling

**Romann M. Weber**                                     *romann.weber@disneyresearch.com*
*Disney Research: Studios*

**Reviewed on OpenReview:** *https: // openreview. net/ forum? id= dpGSNLUCzu*

## Abstract

Implicit generative modeling (IGM) aims to produce samples of synthetic data matching the characteristics of a target data distribution. Recent work (e.g. score-matching networks, diffusion models) has approached the IGM problem from the perspective of pushing synthetic source data toward the target distribution via dynamical perturbations or flows in the ambient space. In this direction, we present the *score difference* (SD) between arbitrary target and source distributions as a flow that optimally reduces the Kullback-Leibler divergence between them while also solving the *Schrödinger bridge* problem. We apply the SD flow to convenient proxy distributions, which are aligned if and only if the original distributions are aligned. We demonstrate the formal equivalence of this formulation to denoising diffusion models under certain conditions. We also show that the training of generative adversarial networks includes a hidden data-optimization sub-problem, which induces the SD flow under certain choices of loss function when the discriminator is optimal. As a result, the SD flow provides a theoretical link between model classes that individually address the three challenges of the *generative modeling trilemma*—high sample quality, mode coverage, and fast sampling—thereby setting the stage for a unified approach.

## 1 Introduction

The goal of implicit generative modeling (IGM) is to create synthetic data samples that are indistinguishable from those drawn from a target distribution. A variety of approaches exist in the literature that address this problem from the perspective of the *dynamics* imposed upon the synthetic data during sampling or training. In particle-optimization sampling methods such as Langevin dynamics (Bussi & Parrinello, 2007; Welling & Teh, 2011), Hamiltonian Monte Carlo (MacKay, 2003), and Stein variational gradient descent (Liu & Wang, 2016), synthetic particles are drawn from a source distribution and perturbed over a number of steps until they resemble particles drawn from the target distribution. Separately, parametric models have been developed that either perform these perturbations implicitly under the hood, as in the case of normalizing flows (Rezende & Mohamed, 2015; Papamakarios et al., 2021), or do so explicitly during inference, as in the case of diffusion models (Ho et al., 2020; Nichol & Dhariwal, 2021).

This would seem to contrast with the training of parametric models for single-step generation by learning a mapping between input drawn from a noise distribution and synthetic output resembling the target data. Perhaps the most famous example of such a model is a generative adversarial network (GAN) (Goodfellow et al., 2014), which leverages a separately trained discriminator to guide the generator toward more target-like output. However, we will show that GAN training contains a hidden sub-problem that induces a flow on the generated data that is completely determined by the loss being optimized. Consequently, this and various other IGM methods can be understood in terms of the dynamics imposed upon the synthetic data.

The question then becomes one of asking what the *optimal* dynamics might be for the IGM task. In this direction, we present the *score difference* (SD)—the difference in the gradients of the log-densities of the target and source data distributions with respect to the data—as the flow direction that optimally reduces the KL divergence between them at each step. We then argue that we can sidestep directly working with the source and target distributions in favor of operating on convenient proxy distributions with common support, which we show are aligned if and only if the original source and target distributions are aligned.

We derive the score difference from the analysis of the dynamical systems that govern probability flow. But we also show that the score difference is hidden within or relates to various other IGM approaches, most notably denoising diffusion models and GANs, under certain conditions. We also outline a flexible algorithmic approach for leveraging SD flow when working with *any* source and target distributions, with no restrictions placed on either distribution.

Our aim is to provide an accessible treatment of a complex topic with important connections to various areas of generative modeling and variational inference. The paper is organized around its key contributions as follows:

1. In Section 2, we derive the score-difference (SD) flow from the study of *probability flow* dynamical systems and show that SD flow optimally reduces the Kullback-Leibler (KL) divergence between the source and target distributions and solves the *Schrödinger bridge problem*.

2. In Section 3, we consider modified proxy distributions for the source and target distributions, which have common support and are generally easier to manage and estimate than the unmodified distributions. We outline a method for aligning these proxies and show that this alignment occurs if and only if the unmodified distributions are aligned.

3. In Section 4, we draw a connection between SD flow and denoising diffusion models and show that they are equivalent under certain conditions. However, unlike diffusion models, SD flow places no restrictions on the prior distribution.

4. In Section 5, we show that GAN generator training is composed of two sub-problems, a *particle-optimization* step that induces a flow determined by the loss being optimized and a *model-optimization* step, in which the flow-perturbed particles are fit by the generator via regression. We then show that the SD flow is induced in GANs in the particle-optimization step under certain conditions and choices of loss function.

5. In Section 6, we present flexible algorithms for applying SD flow to both direct sampling (*particle optimization*) and parametric generator training (*model optimization*).

6. In Section 7, we report experiments on our kernel-based implementation of SD flow, including comparisons to maximum mean discrepancy (MMD) gradient flow and Stein variational gradient descent (SVGD) under a variety of conditions.

We conclude in Section 8. In the appendices we provide supplemental information, discuss theoretical links to MMD gradient flow and SVGD, and report additional experimental results.

## 2 Probability Flow and the Score Difference

### 2.1 Derivation from Stochastic Differential Equations

Consider data $\boldsymbol{x} \in \mathbb{R}^d$ drawn from a base distribution $q = q_0$. We can describe a dynamical system that perturbs the data and evolves its distribution $q_0 \to q_t$ over time by the stochastic differential equation

$$\mathrm{d}\boldsymbol{x} = \boldsymbol{\mu}(\boldsymbol{x}, t)\mathrm{d}t + \sigma(t)\mathrm{d}\boldsymbol{\omega}, \tag{1}$$

where $\boldsymbol{\mu} : \mathbb{R}^d \times \mathbb{R} \to \mathbb{R}^d$ is a *drift coefficient*, $\sigma(t)$ is a *diffusion coefficient*, and $\mathrm{d}\boldsymbol{\omega}$ denotes the standard Wiener process (Song et al., 2020).

When $\boldsymbol{\mu}(\boldsymbol{x}, t) = \frac{\sigma(t)^2}{2}\nabla_{\boldsymbol{x}} \log p(\boldsymbol{x})$, the diffusion in equation 1 describes *Langevin dynamics*, which for a suitable decreasing noise schedule $\sigma(t)$ can be shown to produce samples from a target distribution $p$ as $t \to \infty$ (Bussi & Parrinello, 2007; Welling & Teh, 2011). We indicate this convergence by writing $q_t \rightsquigarrow p$.

As the data points $\boldsymbol{x}$ are perturbed over time $t$, the distribution $q_0$ evolves to $q_t$. This evolution is described by the *Fokker-Planck equation* (Risken, 1996),

$$
\begin{aligned}
\frac{\partial q_t(\boldsymbol{x})}{\partial t} &= -\sum_{i=1}^{d} \frac{\partial}{\partial x_i}[\mu_i(\boldsymbol{x}, t) q_t(\boldsymbol{x})] + \sum_{i=1}^{d}\sum_{j=1}^{d} \frac{\partial^2}{\partial x_i \partial x_j}\left[D_{ij}(\boldsymbol{x}, t) q_t(\boldsymbol{x})\right] \\
&= -\nabla_{\boldsymbol{x}} \cdot [\mu_i(\boldsymbol{x}, t) q_t(\boldsymbol{x})] + \frac{\sigma(t)^2}{2}\nabla_{\boldsymbol{x}}^2 q_t(\boldsymbol{x}) \\
&= -\frac{\sigma(t)^2}{2}\nabla_{\boldsymbol{x}} \cdot [q_t(\boldsymbol{x})\nabla_{\boldsymbol{x}} \log p(\boldsymbol{x}) - \nabla_{\boldsymbol{x}} q_t(\boldsymbol{x})] \\
&= -\frac{\sigma(t)^2}{2}\nabla_{\boldsymbol{x}} \cdot [q_t(\boldsymbol{x}) \left(\nabla_{\boldsymbol{x}} \log p(\boldsymbol{x}) - \nabla_{\boldsymbol{x}} \log q_t(\boldsymbol{x})\right)],
\end{aligned}
\tag{2}
$$

where $\boldsymbol{D}(\boldsymbol{x}, t)$ is an isotropic *diffusion tensor* with elements $D_{ij}(\boldsymbol{x}, t) = \frac{\sigma(t)^2}{2}$ if $i = j$ and $D_{ij}(\boldsymbol{x}, t) = 0$ otherwise, $\nabla^2 = \nabla \cdot \nabla$ represents the *Laplacian operator*,[1] and $\sigma(t)$ and $\boldsymbol{\mu}(\boldsymbol{x}, t)$ are defined as above. When $q_t = p$, equation 2 vanishes, and the evolution of $q_t$ stops. When the drift term $\boldsymbol{\mu}(\boldsymbol{x}, t)$ is defined as above, namely as the gradient of a *potential*,[2] Jordan et al. (1998) show that the dynamics in equation 1 prescribe a direction of steepest descent on a free-energy functional with respect to the Wasserstein metric.

A result due to Anderson (1982) shows that the forward dynamics in equation 1 can be *reversed*, effectively undoing the evolution of $q$ to $p$. These reverse dynamics are given by

$$
\mathrm{d}\boldsymbol{x} = \left[\boldsymbol{\mu}(\boldsymbol{x}, t) - \sigma(t)^2 \nabla_{\boldsymbol{x}} \log q_t(\boldsymbol{x})\right] \mathrm{d}t + \sigma(t)\mathrm{d}\hat{\boldsymbol{\omega}},
\tag{3}
$$

where now $\mathrm{d}t$ is a *negative* time step, and $\hat{\boldsymbol{\omega}}$ is a time-reversed Wiener process. The reverse of a diffusion process is therefore another diffusion process.

Remarkably, there is a *deterministic* process with the same marginal densities as those prescribed by equation 3. The corresponding dynamics are given by the *probability flow* ODE (Maoutsa et al., 2020; Song et al., 2020),

$$
\mathrm{d}\boldsymbol{x} = \left[\boldsymbol{\mu}(\boldsymbol{x}, t) - \frac{\sigma(t)^2}{2}\nabla_{\boldsymbol{x}} \log q_t(\boldsymbol{x})\right] \mathrm{d}t.
\tag{4}
$$

If we substitute in the Langevin drift term $\boldsymbol{\mu}(\boldsymbol{x}, t) = \frac{\sigma(t)^2}{2}\nabla_{\boldsymbol{x}} \log p(\boldsymbol{x})$ from above, equation 4 becomes

$$
\mathrm{d}\boldsymbol{x} = \frac{\sigma(t)^2}{2}\left[\nabla_{\boldsymbol{x}} \log p(\boldsymbol{x}) - \nabla_{\boldsymbol{x}} \log q_t(\boldsymbol{x})\right] \mathrm{d}t.
\tag{5}
$$

Since $\mathrm{d}t$ is assumed to be a negative time step in the reverse process, equation 5 as written provides the dynamics of the *forward* process—pushing $q_t$ *toward* the target distribution $p$—when $\mathrm{d}t$ is positive. Equation 5 represents the **score-difference** (SD) flow of a probability distribution $q_t$ evolving toward $p$ (or away from $p$, depending on the sign). Note that this is no longer a diffusion but rather defines a deterministic trajectory.

Combining equation 2 and equation 5 yields

$$
\frac{\partial q_t(\boldsymbol{x})}{\partial t} = -\nabla_{\boldsymbol{x}} \cdot \left[q_t(\boldsymbol{x})\frac{\mathrm{d}\boldsymbol{x}}{\mathrm{d}t}\right].
\tag{6}
$$

This is the *Liouville equation* (Ehrendorfer, 1994; Maoutsa et al., 2020), which describes the evolution of deterministic systems analogously to how the Fokker-Planck equation describes stochastic systems.

---

[1] Many authors denote the Laplacian by $\Delta$, but we have reserved its use for discrete-time differences.
[2] Here the potential is given by $U(\boldsymbol{x}) = -\log p(\boldsymbol{x})$.

## 2.2 SD Flow Optimally Reduces KL Divergence

A continuous flow $\mathrm{d}\boldsymbol{x} = \mathbf{f}(\boldsymbol{x})\mathrm{d}t$ can be approximated by defining a transformation $\boldsymbol{T}(\boldsymbol{x}) = \boldsymbol{x} + \varepsilon\mathbf{f}(\boldsymbol{x})$ for some small step $\varepsilon > 0$.[3] If $\boldsymbol{x} \sim q$ and $\boldsymbol{x}' = \boldsymbol{T}(\boldsymbol{x}) \sim q_{[\boldsymbol{T}]}$, then Liu & Wang (2016) show that the Kullback-Leibler (KL) divergence between $q_{[\boldsymbol{T}]}$ and $p$,

$$\mathbb{D}_{\mathrm{KL}}(q_{[\boldsymbol{T}]}\|p) = \mathbb{E}_{\boldsymbol{x}\sim q_{[\boldsymbol{T}]}}\left[\log q_{[\boldsymbol{T}]}(\boldsymbol{x}) - \log p(\boldsymbol{x})\right], \tag{7}$$

varies according to its functional derivative,

$$\nabla_\varepsilon\mathbb{D}_{\mathrm{KL}}(q_{[\boldsymbol{T}]}\|p)|_{\varepsilon=0} = -\mathbb{E}_{\boldsymbol{x}\sim q_{[\boldsymbol{T}]}}\left[\mathrm{Tr}\left(\mathcal{A}_p\mathbf{f}(\boldsymbol{x})\right)\right], \tag{8}$$

where

$$\mathcal{A}_p\mathbf{f}(\boldsymbol{x}) = \nabla_{\boldsymbol{x}}\log p(\boldsymbol{x})\mathbf{f}(\boldsymbol{x})^\top + \nabla_{\boldsymbol{x}}\mathbf{f}(\boldsymbol{x}) \tag{9}$$

is the *Stein operator* (Gorham & Mackey, 2015).

By applying Stein's identity (Stein, 1981; Lin et al., 2019) to equation 8, we obtain

$$\begin{aligned}
\mathbb{E}_{\boldsymbol{x}\sim q_{[\boldsymbol{T}]}}\left[\mathrm{Tr}\left(\mathcal{A}_p\mathbf{f}(\boldsymbol{x})\right)\right] &= \mathbb{E}_{\boldsymbol{x}\sim q_{[\boldsymbol{T}]}}\left[\nabla_{\boldsymbol{x}}\log p(\boldsymbol{x})^\top\mathbf{f}(\boldsymbol{x})\right] - \mathbb{E}_{\boldsymbol{x}\sim q_{[\boldsymbol{T}]}}\left[\nabla_{\boldsymbol{x}}\log q_{[\boldsymbol{T}]}(\boldsymbol{x})^\top\mathbf{f}(\boldsymbol{x})\right] \\
&= \mathbb{E}_{\boldsymbol{x}\sim q_{[\boldsymbol{T}]}}\left[\left(\nabla_{\boldsymbol{x}}\log p(\boldsymbol{x}) - \nabla_{\boldsymbol{x}}\log q_{[\boldsymbol{T}]}(\boldsymbol{x})\right)^\top\mathbf{f}(\boldsymbol{x})\right],
\end{aligned} \tag{10}$$

which is the inner product of the score difference and the flow vector $\mathbf{f}(\boldsymbol{x})$.[4] Maximizing the *reduction* in the KL divergence (equation 8) corresponds to maximizing this inner product. Since the inner product of two vectors is maximized when they are parallel, choosing $\mathbf{f}(\boldsymbol{x})$ to output a vector parallel to the score difference will decrease the KL divergence as fast as possible. We can also see from equation 8 and equation 10 that the decrease in the KL divergence is then proportional to the *Fisher divergence*,

$$\mathbb{D}_{\mathrm{F}}(q_{[\boldsymbol{T}]}\|p) = \mathbb{E}_{\boldsymbol{x}\sim q_{[\boldsymbol{T}]}}\left[\|\nabla_{\boldsymbol{x}}\log p(\boldsymbol{x}) - \nabla_{\boldsymbol{x}}\log q_{[\boldsymbol{T}]}(\boldsymbol{x})\|^2\right], \tag{11}$$

which, never being negative, shows that moving along the SD flow in sufficiently small steps will not increase the KL divergence.

We note that the SD flow also naturally emerges from *variational gradient flows* defined over $f$-divergences (Gao et al., 2019; Feng et al., 2021; Gao et al., 2022). Specifically, Gao et al. (2019) show that the negative gradient of the *first variation* of the $f$-divergence functional evaluated at $q$ defines a flow that minimizes the divergence. When that $f$-divergence is the KL divergence, $\mathbb{D}_{\mathrm{KL}}(q\|p)$, the first variation is given by $\delta\mathcal{F}[q]/\delta q = \log q(\boldsymbol{x}) - \log p(\boldsymbol{x}) + 1$, whose negative gradient is the SD flow. Dynamics closely resembling the SD flow also emerge in the study of optimal interventions for constraining stochastic interacting particle systems in *Kullback-Leibler control* problems (Maoutsa & Opper, 2022).

## 2.3 SD Flow Solves the Schrödinger Bridge Problem

The Schrödinger bridge (SB) problem considers the most likely evolution between two marginal densities $q$ and $p$ over time $t \in [0, T]$ (Chen et al., 2015). Specifically, the problem solves

$$P^* = \arg\min_{P\in\Pi(q,p)}\mathbb{D}_{\mathrm{KL}}(P\|Q), \tag{12}$$

where $\Pi(q,p)$ is the collection of densities with marginals $P_0 = q$ and $P_T = p$, and $Q$ is a reference diffusion with initial density $q$ that causes the solution to be unique (Winkler et al., 2023).[5]

Chen et al. (2021) show that solutions to the SB problem correspond to the following forward and backward SDEs:

$$\mathrm{d}\boldsymbol{x} = [\boldsymbol{\mu}(\boldsymbol{x},t) + \sigma^2(t)\nabla\log\Psi(x,t)]dt + \sigma(t)d\boldsymbol{\omega} \tag{13}$$

$$\mathrm{d}\boldsymbol{x} = [\boldsymbol{\mu}(\boldsymbol{x},t) - \sigma^2(t)\nabla\log\hat{\Psi}(\boldsymbol{x},t)]dt + \sigma(t)d\hat{\boldsymbol{\omega}}, \tag{14}$$

---

[3]If $\varepsilon$ is sufficiently small, then the Jacobian of $\boldsymbol{T}$ is of full rank, meaning that the transformation is bijective.

[4]We assume the mild condition that $\mathbf{f}(\boldsymbol{x})p(\boldsymbol{x}) \to \mathbf{0}$ and $\mathbf{f}(\boldsymbol{x})q_{[\boldsymbol{T}]}(\boldsymbol{x}) \to \mathbf{0}$ as $\|\boldsymbol{x}\| \to \infty$.

[5]Note that we are considering the diffusion from $q$ to $p$ for consistency with the Langevin example in Section 2.1.

where $\Psi$ and $\hat{\Psi}$ are non-negative potentials (or *Schrödinger factors*) such that $\Psi(\boldsymbol{x}, t)\hat{\Psi}(\boldsymbol{x}, t) = q_t(\boldsymbol{x})$, and all other notation is as defined in Section 2.1. If we set $\Psi(\boldsymbol{x}, t) = 1$ and $\hat{\Psi}(\boldsymbol{x}, t) = q_t(\boldsymbol{x})$, then the necessary conditions on the potentials are met, and the backward SDE (14) matches equation 3, from which the probability flow ODE in equation 4 and the SD flow in equation 5 directly follow. Therefore, SD flow defines a Schrödinger bridge between the source and target distributions $q$ and $p$.

## 3 Applying SD Flow to Proxy Distributions

One of the difficulties in applying Langevin dynamics or other score-based methods is the requirement that we have access to the true score of the target distribution, $\nabla_{\boldsymbol{x}} \log p(\boldsymbol{x})$, which is almost never available in practice. It is also the case that when operating in the ambient space of $\boldsymbol{x} \in \mathbb{R}^d$, the score may not be well defined in areas of limited support if the data exist on a lower-dimensional manifold, which is generally assumed for a variety of data types of interest, such as image data. A large literature has emerged that is dedicated to the estimation of this score or the design of training procedures that are equivalent to estimating it (Hyvärinen & Dayan, 2005; Song & Ermon, 2020; Song & Kingma, 2021; Karras et al., 2022).

Applying SD flow would appear to be at least twice as difficult, since instead of one score to estimate, now we have two. The distribution $q_t$ is also changing over time, so even a reasonably good estimate at one time would have to be discarded and re-estimated at another. Our approach will be to essentially ignore $p$ and $q_t$ and work instead with modified proxy distributions that are easier to estimate and manage. Importantly, aligning these proxy distributions will automatically align the unmodified source and target distributions.

### 3.1 Aligning Proxy Distributions

We can assess the alignment of two distributions $q$ and $p$ by computing a *statistical distance* between them.[6] Although this quantity is not always a true "distance" in a strict mathematical sense, it will have the two key properties that (1) $\mathbb{D}(q\|p) \geq 0$ for all distributions $p, q$ and (2) $\mathbb{D}(q\|p) = \mathbb{D}(p\|q) = 0$ if and only if $p = q$. Perhaps the best-known statistical distance is the KL divergence, $\mathbb{D}_{\mathrm{KL}}(q\|p)$ (equation 7), although it can diverge to infinity if $p$ and $q$ have unequal support.

One way to equalize the support of two distributions is to corrupt their data with additive noise defined over the whole space $\mathbb{R}^d$. Let us assume a Gaussian noise model. The distribution of $\boldsymbol{z} = \boldsymbol{x} + \sigma\boldsymbol{\epsilon}$, with $\boldsymbol{x} \sim p$ and $\boldsymbol{\epsilon} \sim \mathcal{N}(\boldsymbol{0}, \boldsymbol{I})$, is given by the convolution $\tilde{p} = p * \mathcal{N}(\boldsymbol{0}, \sigma^2\boldsymbol{I})$:

$$\begin{aligned}
\tilde{p}(\boldsymbol{z}) = p(\boldsymbol{z}; \sigma) &= \int_{\mathbb{R}^d} p(\boldsymbol{x})\mathcal{N}(\boldsymbol{z}; \boldsymbol{x}, \sigma^2\boldsymbol{I}) \, \mathrm{d}\boldsymbol{x} \\
&= \mathbb{E}_{\boldsymbol{x}\sim p}\left[\mathcal{N}(\boldsymbol{z}; \boldsymbol{x}, \sigma^2\boldsymbol{I})\right],
\end{aligned} \tag{15}$$

with $\tilde{q}(\boldsymbol{z}) = q(\boldsymbol{z}; \sigma) = \mathbb{E}_{\boldsymbol{y}\sim q}\left[\mathcal{N}(\boldsymbol{z}; \boldsymbol{y}, \sigma^2\boldsymbol{I})\right]$ defined analogously. Although $\mathbb{D}_{\mathrm{KL}}(\tilde{q}\|\tilde{p}) \leq \mathbb{D}_{\mathrm{KL}}(q\|p)$ and $\mathbb{D}_{\mathrm{KL}}(\tilde{q}\|\tilde{p}) \to 0$ as $\sigma \to \infty$ (Sriperumbudur et al., 2017), it is easy to show that $\mathbb{D}_{\mathrm{KL}}(\tilde{q}\|\tilde{p}) = 0$ if and only if $q = p$ (Zhang et al., 2020). As a result, aligning the proxy distributions $\tilde{q}$ and $\tilde{p}$ is equivalent to aligning the generative distribution $q$ with the target distribution $p$.

We have shown that moving parallel to the score difference optimally reduces the KL divergence, so we define an SD flow between $\tilde{q}$ and $\tilde{p}$ to align $q$ with $p$. The score corresponding to $\tilde{p}(\boldsymbol{z}) = p(\boldsymbol{z}; \sigma)$ (15) is

$$\begin{aligned}
\nabla_{\boldsymbol{z}} \log p(\boldsymbol{z}; \sigma) &= \frac{\nabla_{\boldsymbol{z}} p(\boldsymbol{z}; \sigma)}{p(\boldsymbol{z}; \sigma)} \\
&= \frac{\mathbb{E}_{\boldsymbol{x}\sim p}\left[\nabla_{\boldsymbol{z}}\mathcal{N}(\boldsymbol{z}; \boldsymbol{x}, \sigma^2\boldsymbol{I})\right]}{\mathbb{E}_{\boldsymbol{x}\sim p}\left[\mathcal{N}(\boldsymbol{z}; \boldsymbol{x}, \sigma^2\boldsymbol{I})\right]} \\
&= \frac{1}{\sigma^2}\left(\frac{\mathbb{E}_{\boldsymbol{x}\sim p}\left[\mathcal{N}(\boldsymbol{z}; \boldsymbol{x}, \sigma^2\boldsymbol{I})\boldsymbol{x}\right]}{\mathbb{E}_{\boldsymbol{x}\sim p}\left[\mathcal{N}(\boldsymbol{z}; \boldsymbol{x}, \sigma^2\boldsymbol{I})\right]} - \boldsymbol{z}\right).
\end{aligned} \tag{16}$$

---

[6]We will suppress the time index on $q$ here for convenience.

The score for $\tilde{q}(\boldsymbol{z}) = q(\boldsymbol{z}; \sigma)$ is derived in the same way for $\boldsymbol{z} = \boldsymbol{y} + \sigma\boldsymbol{\epsilon}$, with $\boldsymbol{y} \sim q$. This leads to the following expression for the score difference:

$$\nabla_{\boldsymbol{z}} \log p(\boldsymbol{z}; \sigma) - \nabla_{\boldsymbol{z}} \log q(\boldsymbol{z}; \sigma) = \frac{1}{\sigma^2} \left( \frac{\mathbb{E}_{\boldsymbol{x} \sim p}\left[K_\sigma(\boldsymbol{z}, \boldsymbol{x})\boldsymbol{x}\right]}{\mathbb{E}_{\boldsymbol{x} \sim p}\left[K_\sigma(\boldsymbol{z}, \boldsymbol{x})\right]} - \frac{\mathbb{E}_{\boldsymbol{y} \sim q}\left[K_\sigma(\boldsymbol{z}, \boldsymbol{y})\boldsymbol{y}\right]}{\mathbb{E}_{\boldsymbol{y} \sim q}\left[K_\sigma(\boldsymbol{z}, \boldsymbol{y})\right]} \right), \tag{17}$$

where $K_\sigma(\boldsymbol{z}, \boldsymbol{x}) = \exp\left(-\frac{\|\boldsymbol{z}-\boldsymbol{x}\|^2}{2\sigma^2}\right)$ is the Gaussian kernel.[7]

If we set the noise level according to the schedule $\sigma(t) = \sigma$, then the variance term cancels from equation 5, leading to the update

$$\Delta \boldsymbol{z} = \frac{\eta}{2} \left[ \frac{\mathbb{E}_{\boldsymbol{x} \sim p}\left[K_\sigma(\boldsymbol{z}, \boldsymbol{x})\boldsymbol{x}\right]}{\mathbb{E}_{\boldsymbol{x} \sim p}\left[K_\sigma(\boldsymbol{z}, \boldsymbol{x})\right]} - \frac{\mathbb{E}_{\boldsymbol{y} \sim q}\left[K_\sigma(\boldsymbol{z}, \boldsymbol{y})\boldsymbol{y}\right]}{\mathbb{E}_{\boldsymbol{y} \sim q}\left[K_\sigma(\boldsymbol{z}, \boldsymbol{y})\right]} \right] \tag{18}$$

for some $\eta > 0$ defining the step size.

We treat the dynamics of $\boldsymbol{y} \sim q(\boldsymbol{y})$ as being the same as the dynamics of $\boldsymbol{z} \sim \tilde{q}(\boldsymbol{z})$ and set $\Delta \boldsymbol{y} = \Delta \boldsymbol{z}$ under the following rationale: At time $t$, we draw $\boldsymbol{y}_t \sim q_t(\boldsymbol{y})$ and $\boldsymbol{\epsilon} \sim \mathcal{N}(\boldsymbol{0}, \boldsymbol{I})$ to form $\boldsymbol{y}_t + \sigma\boldsymbol{\epsilon} = \boldsymbol{z}_t \sim \tilde{q}(\boldsymbol{z})$. We then perturb $\boldsymbol{z}_t$ according to equation 18, forming $\boldsymbol{z}_t + \Delta \boldsymbol{z}_t = \boldsymbol{z}_{t+1} \sim \tilde{q}_{t+1}(\boldsymbol{z})$, which is closer to the distribution $\tilde{p}(\boldsymbol{z})$ than the point $\boldsymbol{z}_t$ was. However, there is a second way to obtain the point $\boldsymbol{z}_{t+1} \sim \tilde{q}_{t+1}(\boldsymbol{z})$, which is to first create $\boldsymbol{y}_t + \Delta \boldsymbol{y}_t = \boldsymbol{y}_{t+1} \sim q_{t+1}(\boldsymbol{y})$ via the (unknown) update $\Delta \boldsymbol{y}_t$ and then corrupt it with noise to form $\boldsymbol{y}_{t+1} + \sigma\boldsymbol{\epsilon} = \boldsymbol{z}_{t+1} \sim \tilde{q}_{t+1}(\boldsymbol{z})$.[8] If we think of our random noise $\boldsymbol{\epsilon}$ as coming from a queue,[9] then the value of $\boldsymbol{\epsilon}$ is the *same* in both scenarios. The only difference is that $\boldsymbol{\epsilon}$ is drawn from the queue at time $t$ in the first scenario and is saved until time $t + 1$ in the second. As a result, we can equate the two definitions of $\boldsymbol{z}_{t+1}$ and write

$$\overbrace{\underbrace{\boldsymbol{y}_t + \Delta \boldsymbol{y}_t}_{\boldsymbol{y}_{t+1}} + \sigma\boldsymbol{\epsilon}}^{\boldsymbol{z}_{t+1}} = \overbrace{\underbrace{\boldsymbol{y}_t + \sigma\boldsymbol{\epsilon}}_{\boldsymbol{z}_t} + \Delta \boldsymbol{z}_t}^{\boldsymbol{z}_{t+1}}, \tag{19}$$

which, when we cancel like terms, leads to $\Delta \boldsymbol{y}_t = \Delta \boldsymbol{z}_t$. In this case the update $\Delta \boldsymbol{y}$ does not necessarily equal the score difference of the "clean" distributions, $\nabla_{\boldsymbol{y}} \log p(\boldsymbol{y}) - \nabla_{\boldsymbol{y}} \log q(\boldsymbol{y})$, but is rather the perturbation to the clean data required for optimally aligning the proxies $\tilde{q}$ and $\tilde{p}$. Recalling that the alignment of the proxy distributions implies the alignment of the original distributions, the update $\Delta \boldsymbol{y}$ still serves to align $q$ with $p$.

## 3.2 Limitations and Alternative Formulations of SD Flow

In the limit of infinite data, equation 17 is exact. But applying this formulation in a large-data setting can be computationally expensive, and estimates using smaller batches may suffer from low accuracy, especially in high dimensions. Recent work by Ba et al. (2021) shows that SVGD, which has a kernel-based specification similar to our description of SD flow in Section 3.1, tends to underestimate the target variance unless the number of particles is greater than the dimensionality. However, those authors note that MMD gradient descent (Arbel et al., 2019), which is closer in formulation to SD flow (see Appendix B.3), does not have this limitation. Additionally, recent work by Wang et al. (2022) proposed a Wasserstein gradient descent method essentially identical to the SD flow formulation in Section 3.1, albeit operating in a projected space, which the authors claim scales favorably to high dimensions.

In any case, it is useful to consider each term of equation 17 as a *module* that can be swapped out for another estimate, depending on the problem setting. For example, we can rewrite equation 17 in the equivalent form

$$\nabla_{\boldsymbol{z}} \log p(\boldsymbol{z}; \sigma) - \nabla_{\boldsymbol{z}} \log q(\boldsymbol{z}; \sigma) = \frac{1}{\sigma^2} \left[ \mathbb{E}[\boldsymbol{x}|\boldsymbol{z}] - \mathbb{E}[\boldsymbol{y}|\boldsymbol{z}] \right] \tag{20}$$

$$= \frac{1}{\sigma^2} \left[ D_p^*(\boldsymbol{z}; \sigma) - D_{q_t}^*(\boldsymbol{z}; \sigma) \right], \tag{21}$$

where $D_p^*(\boldsymbol{z}; \sigma)$ and $D_{q_t}^*(\boldsymbol{z}; \sigma)$ are the *optimal* denoising models for the distributions $p$ and $q_t$, respectively, when corrupted by Gaussian noise at level $\sigma$. A simple derivation of this result is possible by rearranging

---

[7]This is possible because the normalization constant of the normal distribution cancels from the numerator and denominator.
[8]Here we assume a constant noise schedule, $\sigma(t) = \sigma$ for all $t$.
[9]This is the case with random number tables as well as our computers' pseudorandom number generators.

Tweedie's formula (Efron, 2011),[10] but we provide a separate proof of optimality in Appendix B.1. This formulation is particularly well-suited to high-dimensional applications, as it admits the use of specialized U-net architectures that form the backbone of modern denoising diffusion models (Karras et al., 2022).

As a practical consideration, the denoiser corresponding to the target data would need to be trained only once, while the denoiser for the generative distribution would, at least in principle, need to be retrained after each step along the flow. This is not in itself a major limitation, since the iterative fine-tuning of the generative-distribution denoiser $D_{q_t}$ is no more onerous than the standard practice of training a GAN generator and discriminator in alternating steps, although it does create a potential burden on resources by requiring a second denoiser to be loaded in memory. However, since we actually observe $\boldsymbol{y} \sim q_t$ before corrupting it to form $\boldsymbol{z} = \boldsymbol{y} + \sigma \boldsymbol{\epsilon}$, we can replace $D_{q_t}^*(\boldsymbol{z}; \sigma) = \mathbb{E}[\boldsymbol{y}|\boldsymbol{z}]$ with $\boldsymbol{y}$ in equation 21,[11] leading to the update

$$\boldsymbol{y} \leftarrow (1 - \rho)\boldsymbol{y} + \rho D_p^*(\boldsymbol{z}; \sigma) \tag{22}$$

for some small step size $\rho$. In Section 4, we show that this approximate formulation of SD flow is equivalent to the reverse process in denoising diffusion models.

## 4    Relation to Denoising Diffusion Models

In diffusion modeling, data from the target distribution $p$ is corrupted in a forward diffusion process by Gaussian noise under the scale and noise schedules $\alpha_t = \alpha(t)$ and $\sigma_t = \sigma(t)$, respectively. Then for $\boldsymbol{z}_0 = \boldsymbol{x} \sim p$, the conditional distribution at time $t$ relative to that at time $s < t$ is given by

$$p(\boldsymbol{z}_t|\boldsymbol{z}_s) = \mathcal{N}(\alpha_{t|s}\boldsymbol{z}_s, \sigma_{t|s}^2 \boldsymbol{I}),$$

where $\alpha_{t|s} = \alpha_t/\alpha_s$ and $\sigma_{t|s}^2 = \sigma_t^2 - \alpha_{t|s}^2 \sigma_s^2$ (Kingma et al., 2021).

The hard part is inferring the *reverse* diffusion process, $p(\boldsymbol{z}_s|\boldsymbol{z}_t)$, which is intractable unless also conditioned on $\boldsymbol{z}_0 = \boldsymbol{x}$: $p(\boldsymbol{z}_s|\boldsymbol{z}_t, \boldsymbol{x}) = \mathcal{N}(\boldsymbol{\mu}_{s|t}, \sigma_{s|t}^2)$, where $\boldsymbol{\mu}_{s|t} = (\alpha_{t|s}\sigma_s^2/\sigma_t^2)\boldsymbol{z}_t + (\alpha_s \sigma_{t|s}^2/\sigma_t^2)\boldsymbol{x}$ and $\sigma_{s|t}^2 = \sigma_{t|s}^2 \sigma_s^2/\sigma_t^2$. In practice, $\boldsymbol{x}$ is replaced by $D(\boldsymbol{z}_t; \sigma_t)$, the output of a denoising model.[12]

If we let $\alpha_s = \alpha_t = 1$ for all $s, t$, then

$$\boldsymbol{z}_s = \frac{\sigma_s^2}{\sigma_t^2}\boldsymbol{z}_t + \left(1 - \frac{\sigma_s^2}{\sigma_t^2}\right)D(\boldsymbol{z}_t; \sigma_t) + \sigma_{s|t}\boldsymbol{\epsilon}_s \tag{23}$$

$$= (1 - \rho)\boldsymbol{z}_t + \rho D(\boldsymbol{z}_t; \sigma_t) + \sqrt{\rho}\sigma_s\boldsymbol{\epsilon}_s \tag{24}$$

for $\boldsymbol{\epsilon}_s \sim \mathcal{N}(\boldsymbol{0}, \boldsymbol{I})$ and $\rho = 1 - \sigma_s^2/\sigma_t^2$. Recalling that in our framework, $\boldsymbol{z}_t = \boldsymbol{y}_t + \sigma_t\boldsymbol{\epsilon}_t \sim q_t(\boldsymbol{z}_t; \sigma_t)$ for all $t$, equation 24 becomes

$$
\begin{aligned}
\boldsymbol{z}_s &= (1 - \rho)\boldsymbol{y}_t + \rho D(\boldsymbol{z}_t; \sigma_t) + \sqrt{\rho}\sigma_s\boldsymbol{\epsilon}_s + (1 - \rho)\sigma_t\boldsymbol{\epsilon}_t \\
&= (1 - \rho)\boldsymbol{y}_t + \rho D(\boldsymbol{z}_t; \sigma_t) + \sqrt{\rho\sigma_s^2 + (1 - \rho)^2\sigma_t^2}\,\boldsymbol{\epsilon} \\
&= \underbrace{(1 - \rho)\boldsymbol{y}_t + \rho D(\boldsymbol{z}_t; \sigma_t)}_{\boldsymbol{y}_s} + \underbrace{\sqrt{\left(1 - \frac{\sigma_s^2}{\sigma_t^2}\right)\sigma_s^2 + \left(\frac{\sigma_s^2}{\sigma_t^2}\right)^2 \sigma_t^2}}_{\sigma_s}\,\boldsymbol{\epsilon} \\
&= \boldsymbol{y}_s + \sigma_s\boldsymbol{\epsilon},
\end{aligned}
\tag{25}
$$

where $\boldsymbol{\epsilon}, \boldsymbol{\epsilon}_s, \boldsymbol{\epsilon}_t \sim \mathcal{N}(\boldsymbol{0}, \boldsymbol{I})$ and $\boldsymbol{y}_s$ follows from equation 22. In other words, the updating process under SD flow is equivalent to the denoising diffusion reverse process under the substitution described in Section 3.2.

---

[10]Tweedie's formula states that $\mathbb{E}[\boldsymbol{x}|\boldsymbol{z}] = \boldsymbol{z} + \sigma^2 \nabla_{\boldsymbol{z}} \log p(\boldsymbol{z}; \sigma)$.

[11]This is an approximation, since $D_{q_t}^*(\boldsymbol{z}; \sigma) = \mathbb{E}[\boldsymbol{y}|\boldsymbol{z}]$ will not necessarily equal $\boldsymbol{y}$ and may be closer to a local mean for large $\sigma$.

[12]In alternative but equivalent implementations, the error between $\boldsymbol{x}$ and $\boldsymbol{z}_t$ is predicted by a parametric model $\boldsymbol{\epsilon}_{\boldsymbol{\theta}}(\boldsymbol{z}_t; t)$.

## 5 Implicit Flows in Generative Adversarial Networks

### 5.1 Decomposing Generator Training into Sub-problems

When training a generative model $g_{\boldsymbol{\theta}}$, we define a loss $\mathcal{L}$, which is a scalar function that quantifies the discrepancy between the current model output and the target distribution. We typically treat this loss as a function of the parameters $\boldsymbol{\theta} \in \mathbb{R}^n$ and then optimize $\boldsymbol{\theta}$ to minimize $\mathcal{L}$ via gradient[13] descent at some learning rate $\eta > 0$:

$$\boldsymbol{\theta}' = \boldsymbol{\theta} - \eta \left(\frac{\partial \mathcal{L}}{\partial \boldsymbol{\theta}}\right)^{\top}. \tag{26}$$

However, the loss $\mathcal{L}$ is also a function of the generated data $\boldsymbol{y} = g_{\boldsymbol{\theta}}(\boldsymbol{\xi}) \in \mathbb{R}^d$, which is *itself* a function of either $\boldsymbol{\xi} \in \mathbb{R}^l$ or $\boldsymbol{\theta} \in \mathbb{R}^n$, depending on our perspective. This perspective can be made explicit by decomposing the derivative of the loss via the multivariate chain rule,

$$\underbrace{\frac{\partial \mathcal{L}}{\partial \boldsymbol{\theta}}}_{1 \times n} = \underbrace{\frac{\partial \mathcal{L}}{\partial \boldsymbol{y}}}_{1 \times d} \underbrace{\frac{\partial \boldsymbol{y}}{\partial \boldsymbol{\theta}}}_{d \times n}. \tag{27}$$

This allows us to consider each component of the decomposition as corresponding to its own sub-problem.

In the **first sub-problem**, we perturb the generated data $\boldsymbol{y}$ in the direction of the negative gradient,

$$\boldsymbol{y}' = \boldsymbol{y} - \lambda_1 \left(\frac{\partial \mathcal{L}}{\partial \boldsymbol{y}}\right)^{\top}, \tag{28}$$

where $\lambda_1 > 0$ is some small step size. Intuitively, the perturbed data $\boldsymbol{y}'$ corresponds to a potential output of the generator that would have a lower loss, but we can also interpret it as resulting from a gradient flow on the synthetic data. In GANs, this flow serves to approximately "invert" the discriminator (Weber, 2021).

In the **second sub-problem**, we update the generator parameters $\boldsymbol{\theta}$ by regressing the new, perturbed target $\boldsymbol{y}'$ on the original generator input $\boldsymbol{\xi}$ via the least-squares loss,

$$\mathcal{J} = \frac{1}{2}\|g_{\boldsymbol{\theta}}(\boldsymbol{\xi}) - \boldsymbol{y}'\|^2 = \frac{1}{2}\|\boldsymbol{y} - \boldsymbol{y}'\|^2. \tag{29}$$

Putting the pieces together leads to the parameter update

$$\boldsymbol{\theta}' = \boldsymbol{\theta} - \lambda_2 \left(\frac{\partial \mathcal{J}}{\partial \boldsymbol{\theta}}\right)^{\top} \tag{30}$$

$$= \boldsymbol{\theta} - \lambda_2 \left(\frac{\partial g_{\boldsymbol{\theta}}(\boldsymbol{\xi})}{\partial \boldsymbol{\theta}}\right)^{\top} (g_{\boldsymbol{\theta}}(\boldsymbol{\xi}) - \boldsymbol{y}') \tag{31}$$

$$= \boldsymbol{\theta} - \lambda_2 \left(\frac{\partial \boldsymbol{y}}{\partial \boldsymbol{\theta}}\right)^{\top} (\boldsymbol{y} - \boldsymbol{y}') \tag{32}$$

$$= \boldsymbol{\theta} - \lambda_1 \lambda_2 \left(\frac{\partial \boldsymbol{y}}{\partial \boldsymbol{\theta}}\right)^{\top} \left(\frac{\partial \mathcal{L}}{\partial \boldsymbol{y}}\right)^{\top} \tag{33}$$

$$= \boldsymbol{\theta} - \lambda_1 \lambda_2 \left(\frac{\partial \mathcal{L}}{\partial \boldsymbol{\theta}}\right)^{\top}, \tag{34}$$

which is is equal to the standard gradient update of $\boldsymbol{\theta}$ under the original loss $\mathcal{L}$ (equation 26) with step size $\eta = \lambda_1 \lambda_2$. Here equation 33 follows from equation 32 by rearranging and substituting equation 28, while equation 34 follows from equation 33 via equation 27.

---

[13]We treat the gradient as the transpose of the derivative.

Although this decomposition is a direct consequence of gradient descent, it shows that hidden within generator training are two sub-problems with separate control options (their learning rates, for instance), each of which may be easier to conceptualize and handle than the original problem. In particular, we see that the model-optimization step of generator training is preceded, at least implicitly, by a particle-optimization step that prescribes a flow in the ambient data space $\mathbb{R}^d$, regardless of the overall loss being optimized. This suggests that we can treat this particle-optimization step as a *target-generation* module that can be swapped out in favor of other procedures, such as SD flow. We note that this interpretation is consistent with recently proposed flow-based methods for training parametric generators (Gao et al., 2019; 2022). Furthermore, it suggests that a wide variety of generative models can be understood in terms of the dynamics imposed on the generated data during training.

## 5.2 SD Flow in GANs

Many GAN generators employ the widely used *non-saturating loss* (Goodfellow, 2016) given by

$$\mathcal{L}(\boldsymbol{\theta}) = -\mathbb{E}_{\boldsymbol{\xi} \sim p_0} \left[ \log f(g_{\boldsymbol{\theta}}(\boldsymbol{\xi})) \right] \approx -\frac{1}{|\mathcal{B}|} \sum_{\boldsymbol{y} \in \mathcal{B} \sim q_t} \log f(\boldsymbol{y}), \tag{35}$$

where $\boldsymbol{\xi} \in \mathbb{R}^l$ is a random noise input to the generator drawn from a prior distribution $p_0$ and $f : \mathbb{R}^d \to (0, 1)$ is a separately trained discriminator that estimates the probability that its argument is real data coming from a target distribution $p$ (in which case $f \approx 1$), as opposed to fake data coming from the generator distribution $q_t$ (in which case $f \approx 0$). We note that in every practical case we are working with an empirical estimate of the expectation over a finite batch of generated data $\boldsymbol{y} = g_{\boldsymbol{\theta}}(\boldsymbol{\xi})$ collected in the set $\mathcal{B}$. Intuitively, the aim of the loss given by equation 35 is to tune the parameters $\boldsymbol{\theta}$ to *maximize* the discriminator's assessment of generated data as real.

It can be shown that, if the prior probabilities of coming from either $p$ or $q_t$ are equal, the *Bayes optimal* classifier $f_t$ is given by

$$f_t(\boldsymbol{y}) = \frac{p(\boldsymbol{y})}{p(\boldsymbol{y}) + q_t(\boldsymbol{y})}, \tag{36}$$

where we have included the time subscript to indicate the optimal discriminator's dependence on the changing distribution $q_t$. An optimal discriminator is often assumed in the analysis of GANs but almost never holds in actual practice.

When implemented as a neural network, the discriminator $f_t$ usually terminates with a sigmoid activation,

$$f_t(\boldsymbol{y}) = \frac{1}{1 + \exp[-h_t(\boldsymbol{y})]}, \tag{37}$$

where $h_t(\boldsymbol{y})$ is the *pre-activation* output of the discriminator $f_t$. Equating equation 36 and equation 37, we see that in the case of an optimal discriminator, $h_t(\boldsymbol{y}) = \log p(\boldsymbol{y}) - \log q_t(\boldsymbol{y})$, whose gradient is the score difference,

$$\nabla_{\boldsymbol{y}} h_t(\boldsymbol{y}) = \nabla_{\boldsymbol{y}} \log p(\boldsymbol{y}) - \nabla_{\boldsymbol{y}} \log q_t(\boldsymbol{y}). \tag{38}$$

When trained using the non-saturating loss (equation 35), the gradient flow induced on a point $\boldsymbol{y}$ is

$$
\begin{aligned}
-\nabla_{\boldsymbol{y}} \mathcal{L} &= \frac{1}{|\mathcal{B}|} \frac{\nabla_{\boldsymbol{y}} f_t(\boldsymbol{y})}{f_t(\boldsymbol{y})} + \frac{1}{|\mathcal{B}|} \sum_{\boldsymbol{y}' \in \mathcal{B} \setminus \boldsymbol{y}} \underbrace{\frac{\nabla_{\boldsymbol{y}} f_t(\boldsymbol{y}')}{f_t(\boldsymbol{y}')}}_{0} \\
&= \frac{1}{|\mathcal{B}|} (1 - f_t(\boldsymbol{y})) \nabla_{\boldsymbol{y}} h_t(\boldsymbol{y}) \\
&\propto (1 - f_t(\boldsymbol{y})) \left[ \nabla_{\boldsymbol{y}} \log p(\boldsymbol{y}) - \nabla_{\boldsymbol{y}} \log q_t(\boldsymbol{y}) \right].
\end{aligned}
\tag{39}
$$

Since $[1 - f_t(\boldsymbol{y})] > 0$ for all $\boldsymbol{y}$, taking the results of Section 5.1 into account, we see that standard GAN training with an optimal discriminator consistently pushes the generated data toward the target data in a direction parallel to the score difference (equation 38).

We can also consider an alternative to the non-saturating loss that focuses on the sum of the discriminator's pre-activation outputs for generated data,

$$\mathcal{L}^{\mathrm{alt}} = -\sum_{\boldsymbol{y} \in \mathcal{B}} h_t(\boldsymbol{y}), \tag{40}$$

which induces a flow exactly equal to the score difference when the discriminator is optimal:

$$-\nabla_{\boldsymbol{y}} \mathcal{L}^{\mathrm{alt}} = \nabla_{\boldsymbol{y}} h_t(\boldsymbol{y}) + \sum_{\boldsymbol{y}' \in \mathcal{B} \setminus \boldsymbol{y}} \underbrace{\nabla_{\boldsymbol{y}} h_t(\boldsymbol{y}')}_{0}$$

$$= \nabla_{\boldsymbol{y}} \log p(\boldsymbol{y}) - \nabla_{\boldsymbol{y}} \log q_t(\boldsymbol{y}). \tag{41}$$

## 6 Algorithms

In this section we provide pseudocode algorithms for two main applications of SD flow: (1) direct sampling via *particle optimization*, which transports a set of particles from a source distribution to match a target distribution, interpolating between the distributions in the process, and (2) the *model optimization* of a parametric generator by (a) progressively perturbing generator output toward the target distribution and then (b) regressing those perturbed targets on the generator input. In Section 5, we showed that the training of a GAN generator can be decomposed into steps (a) and (b). This creates the opportunity to replace a separately trained discriminator with another target-generation method, such as SD flow, in step (a).

The algorithms reflect that SD flow has more than one representation or specification. There is the *kernel-based* specification (Section 3.1) derived from considering noise-injected proxy distributions; there is the *denoiser-based* specification (Section 3.2), which exploits a link between SD flow and diffusion models; and there is the *density-ratio* specification (Section 5) as estimated via a discriminator, such as one would use in GAN training. Other specifications and estimation methods are possible, especially considering the fundamental role played by the density ratio in machine learning applications (Sugiyama et al., 2012).

### 6.1 Particle Optimization

---
**Algorithm 1** Particle optimization with SD flow
---
**Input:** Target data $\{\boldsymbol{x}_i\}_{i=1}^N \sim p$, base (prior) data $\{\boldsymbol{y}_j\}_{j=1}^M \sim q_0$, noise schedule $\sigma(t)$, step schedule $\eta(t)$
**repeat**
    Draw data batches $\boldsymbol{x} \sim p$ and $\boldsymbol{y} \sim q_t$
    Draw noise $\boldsymbol{\epsilon} \sim \mathcal{N}(\mathbf{0}, \boldsymbol{I})$ and perturb data: $\boldsymbol{z} = \boldsymbol{y} + \sigma(t)\boldsymbol{\epsilon}$
    # Calculate SD (equation 18, 21, or 38).
    $\Delta \boldsymbol{z} \propto \frac{\mathbb{E}_{\boldsymbol{x} \sim p}[K_{\sigma(t)}(\boldsymbol{z}, \boldsymbol{x})\boldsymbol{x}]}{\mathbb{E}_{\boldsymbol{x} \sim p}[K_{\sigma(t)}(\boldsymbol{z}, \boldsymbol{x})]} - \frac{\mathbb{E}_{\boldsymbol{y} \sim q_t}[K_{\sigma(t)}(\boldsymbol{z}, \boldsymbol{y})\boldsymbol{y}]}{\mathbb{E}_{\boldsymbol{y} \sim q_t}[K_{\sigma(t)}(\boldsymbol{z}, \boldsymbol{y})]} = D_p^*(\boldsymbol{z}; \sigma(t)) - D_{q_t}^*(\boldsymbol{z}; \sigma(t)) \propto \nabla_{\boldsymbol{z}} h_t(\boldsymbol{z})$
    # Move (clean) data in SD direction.
    $\boldsymbol{y} \leftarrow \boldsymbol{y} + \eta(t)\Delta \boldsymbol{z}$
**until** Terminated
---

In the *particle-optimization* application (Algorithm 1), a sample is generated by perturbing a single batch of "base data," much like one would via Langevin dynamics or Hamiltonian Monte Carlo. Here we interpret the base data as being drawn from $q_0$ and evolving to the distribution $q_t \rightsquigarrow p$ through iterative perturbation.

### 6.2 Model Optimization

Ordinary regression problems involve paired training data $\{(\boldsymbol{\xi}, \boldsymbol{y})\}$ implicitly defining the mapping $g : \boldsymbol{\xi} \mapsto \boldsymbol{y}$ to be learned. In the case of generative modeling, where the aim is to map data from a prior noise distribution to a target distribution, no *a priori* pairing exists. This requires the mapping to be learned indirectly. With GANs, the sub-problem interpretation of Section 5.1 is that the discriminator provides this pairing by associating a noise input $\boldsymbol{\xi}$ with a perturbed output $\boldsymbol{y}'$ that then serves as a regression target.

---

**Algorithm 2** Model optimization with SD flow

---

**Input:** Target data $\{\boldsymbol{x}_i\}_{i=1}^N \sim p$, noise input for generator $\boldsymbol{\xi} \sim p_0$, initialized generator $g_{\boldsymbol{\theta}}$ ($\boldsymbol{\theta} \in \mathbb{R}^n$), noise schedule $\sigma(t)$, step schedule $\eta(t)$, learning rate schedule $\lambda(t)$

**repeat**
     Draw data batches $\boldsymbol{x} \sim p$ and $\boldsymbol{\xi} \sim p_0$
     Generate sample $\boldsymbol{y} = g_\theta(\boldsymbol{\xi})$
     Draw noise $\boldsymbol{\epsilon} \sim \mathcal{N}(\mathbf{0}, \boldsymbol{I})$ and perturb data: $\boldsymbol{z} = \boldsymbol{y} + \sigma(t)\boldsymbol{\epsilon}$
     # Calculate SD (equation 18, 21, or 38).
     $\Delta\boldsymbol{z} \propto \frac{\mathbb{E}_{\boldsymbol{x}\sim p}\left[K_{\sigma(t)}(\boldsymbol{z},\boldsymbol{x})\boldsymbol{x}\right]}{\mathbb{E}_{\boldsymbol{x}\sim p}\left[K_{\sigma(t)}(\boldsymbol{z},\boldsymbol{x})\right]} - \frac{\mathbb{E}_{\boldsymbol{y}\sim q_t}\left[K_{\sigma(t)}(\boldsymbol{z},\boldsymbol{y})\boldsymbol{y}\right]}{\mathbb{E}_{\boldsymbol{y}\sim q_t}\left[K_{\sigma(t)}(\boldsymbol{z},\boldsymbol{y})\right]} = D_p^*(\boldsymbol{z};\sigma(t)) - D_{q_t}^*(\boldsymbol{z};\sigma(t)) \propto \nabla_{\boldsymbol{z}} h_t(\boldsymbol{z})$
     # Move (clean) data in SD direction.
     $\boldsymbol{y} \leftarrow \boldsymbol{y} + \eta(t)\Delta\boldsymbol{z}$
     # Update generator parameters via regression.
     $\boldsymbol{\theta} \leftarrow \boldsymbol{\theta} - \frac{\lambda(t)}{2}\nabla_{\boldsymbol{\theta}}\|g_\theta(\boldsymbol{\xi}) - \boldsymbol{y}\|^2$
**until** Terminated

---

The *model-optimization* application (Algorithm 2) trains a generator with unpaired data via regression on perturbed targets that move progressively closer to the target distribution. Here we interpret $q_0$ as the distribution of the output of the generator $g_{\boldsymbol{\theta}}$ before training. As the generator regresses on perturbed targets, its output distribution changes to $q_t$ at time step $t$ as governed by the Liouville equation.[14] When the choice of SD flow is the gradient of the pre-activation discriminator output, $\nabla_{\boldsymbol{z}} h_t(\boldsymbol{z})$ (equation 38), this algorithm is equivalent to GAN training with the alternative loss described in equation 40.

## 7 Experiments

Here we report several experiments on toy data. We focus in this section on particle-optimization applications using the kernel-based definition of SD flow (Section 3.1). This allows us to compare the performance of SD flow against two other kernel-based particle-optimization algorithms, namely MMD gradient flow (Arbel et al., 2019) and Stein variational gradient descent (SVGD) (Liu & Wang, 2016). Consistent with those works, we focus on low-dimensional data, although we present additional results on moderately high-dimensional data in a model-optimization application in Appendix C.4.

We leave a thorough exploration of our alternative specifications of SD flow (Section 3.2) on high-dimensional image data to follow-up work, since beyond the considerable computational and data demands in that setting, there are many architectural design choices and training tricks behind the current state of the art in image generation, which fall outside the scope of this paper. Nevertheless, several suggestions for applying SD flow to high-dimensional data, such as images, are given in Section 3.2. The general procedures given in Algorithms 1 and 2 still apply in the high-dimensional case.

### 7.1 Experimental Conditions

For the experiments reported in this section, we vary the conditions below, which we describe along with the labels used in Tables 1 and 2. All methods were tested with a default learning rate of $\eta = 0.1$.

**ADAGRAD:**    The published implementation of SVGD (Liu & Wang, 2016) relies on the adaptive gradient optimization algorithm AdaGrad (Duchi et al., 2011). We tested all algorithms using AdaGrad versus standard stochastic gradient descent (SGD).

**BATCH:**    The moderate size of our toy data sets allows us to test both batch-based and full-data versions of the algorithms. Batch sizes of 128 were used in the batch condition.

**CONST:**    In Section 4, we showed that an approximation of SD flow is equivalent to denoising diffusion under a decreasing noise schedule. However, we also showed in Section 3.1 that we can work entirely

---

[14]See Section 2.1 and Appendix B.2.

with noise-corrupted proxy distributions, which allows for a *constant* noise schedule. Both MMD gradient flow and SVGD can be interpreted in the constant-noise setting, so we tested all algorithms with both constant and varying noise schedules. When using a constant noise schedule, we used the method described by Liu & Wang (2016), which determines the kernel bandwidth as a function of the median squared pairwise distance among the source (base) data and the number of points.[15] In the non-constant setting, we used a modified cosine noise schedule to interpolate between a maximum and minimum variance: $\sigma^2(t) = \sigma_{\max}^2 \cos(\pi t/2)$ for $t \in [0, t_{\max}]$, with $t_{\max} = 2/\pi \cos^{-1}(\sigma_{\min}^2/\sigma_{\max}^2)$.

**ANNEAL:** Although the kernel-based realization of SD flow is motivated by considering noise-annealed proxy distributions (Section 3.1), annealing is not necessary for SD flow to converge. Further, since SVGD does not anneal data during training, and MMD gradient flow introduces annealing only as a regularization technique, we tested all algorithms in both annealed and non-annealed conditions.

**OFFSET:** In our experiments in $\mathbb{R}^3$, we introduced the condition of initializing the base distribution either away from the center of the target distribution (offset) or centered at the target distribution's mean.

A discussion of the relationship between SD flow and MMD gradient flow is given in Appendix B.3 and between SD flow and SVGD in Appendix B.4. That discussion provides additional context behind some of the experimental conditions described above.

### 7.2 Results

To measure distribution alignment, we define a mean *characteristic function distance* (CFD),

$$\mathbb{D}_{\mathrm{CF}}(p\|q_t) = \frac{1}{K} \sum_{k=1}^{K} |\mathbb{E}_{\boldsymbol{x} \sim p}\left[\exp(\mathrm{i}\boldsymbol{\omega}_k^\top \boldsymbol{x})\right] - \mathbb{E}_{\boldsymbol{y} \sim q_t}\left[\exp(\mathrm{i}\boldsymbol{\omega}_k^\top \boldsymbol{y})\right]|,$$

where $\mathrm{i} = \sqrt{-1}$ is the imaginary unit. $\mathbb{D}_{\mathrm{CF}}(p\|q_t)$ is the mean absolute difference between the empirical characteristic functions of $p$ and $q_t$ evaluated at $K$ frequencies ($K = 256$ in our case) $\boldsymbol{\omega}_k \in \mathbb{R}^d$ drawn from a normal distribution. For the reporting in Tables 1 and 2, we establish convergence in the following way: We compute the CFD for two independent random samples of the target distribution and record the *maximum* value over 1000 trials, which provides a threshold for the CFD estimated from finite data when the distributions are equal. We say that an algorithm has converged if the CFD between the synthetic and target distributions falls below this threshold.

#### 7.2.1 Fitting a 25-Gaussian Grid

For our first particle-optimization experiment, we created a target distribution of 1024 points drawn from a mixture of 25 spherical Gaussians arranged on a grid in $\mathbb{R}^2$ and initialized 1024 points from a spherical Gaussian base distribution at a large distance from the target distribution but closest to its top-left component. (See Figure 1.) For the non-constant condition, we used a cosine noise schedule (described above) with $\sigma_{\max}^2 = 10$ and $\sigma_{\min}^2 = 0.5$. The results of this experiment are summarized in Table 1. Note that SD flow is the only algorithm to converge in every condition and that each algorithm consistently either converges or fails to converge in each condition.

#### 7.2.2 Fitting a Gaussian Mixture

In this experiment, our target "mystery distribution" $p$ is a mixture of 30 Gaussians in $\mathbb{R}^3$ arranged in the shape of a question mark. (See Figure 2.) In addition to the experimental conditions tested in Section 7.2.1, we also tested initializing the base distribution offset from the target distribution versus centered at the target distribution's mean. For the non-constant condition, we used a cosine noise schedule (described above) with $\sigma_{\max}^2 = 4$ and $\sigma_{\min}^2 = 0.5$. The convergence results are given in Table 2. Once again, SD flow is the only algorithm to converge in every condition.

---

[15] $\sigma^2 = 2 \times \mathrm{median}[D^2(\boldsymbol{y}, \boldsymbol{y}')]/\log(N+1)$

Table 1: Convergence probabilities (final three columns) for the SD flow (SD), MMD gradient flow (MMD) and Stein variational gradient descent (SVGD) algorithms after 1000 iterations over five independent trials on the 25-Gaussian particle-optimization problem in $\mathbb{R}^2$ under the experimental conditions described in Section 7.1. Convergence is defined as obtaining a minimum characteristic function distance less than a threshold empirically determined by comparing independent copies of the target distribution (see text). SD flow is the only algorithm to converge under all conditions.

| ADAGRAD | BATCH | CONST | ANNEAL | SD (OURS) | MMD | SVGD |
|---------|-------|-------|--------|-----------|-----|------|
| N | N | N | N | 1 | 0 | 0 |
| N | N | N | Y | 1 | 0 | 0 |
| N | N | Y | N | 1 | 0 | 0 |
| N | N | Y | Y | 1 | 0 | 0 |
| N | Y | N | N | 1 | 0 | 0 |
| N | Y | N | Y | 1 | 0 | 0 |
| N | Y | Y | N | 1 | 0 | 0 |
| N | Y | Y | Y | 1 | 0 | 0 |
| Y | N | N | N | 1 | 0 | 1 |
| Y | N | N | Y | 1 | 0 | 1 |
| Y | N | Y | N | 1 | 0 | 1 |
| Y | N | Y | Y | 1 | 0 | 1 |
| Y | Y | N | N | 1 | 0 | 1 |
| Y | Y | N | Y | 1 | 0 | 1 |
| Y | Y | Y | N | 1 | 0 | 1 |
| Y | Y | Y | Y | 1 | 0 | 1 |

### 7.2.3 Discussion of Experimental Results

Our experiments show that there are conditions under which all tested algorithms successfully and consistently transport particles from a source distribution to a target distribution. However, only SD flow showed itself to be robust to all experimental conditions. SVGD converged in every condition in which AdaGrad was employed in our experiments, where its performance was extremely close to that of SD flow. However, SVGD consistently failed to converge in any condition in which AdaGrad was not employed, while SD flow did converge in these conditions. MMD gradient flow fared worse in general and exhibited a tendency to cause a subset of the synthetic data to diverge.[16] We note, however, that noise plays a different role in our setup than it does in the work of Arbel et al. (2019), where it serves a regularizing purpose.

The convergence results reported in Tables 1 and 2 were determined by comparing the minimum CFD achieved by an algorithm after 1000 iterations to a threshold empirically determined by multiple comparisons of independent random draws of the target distribution. Failure to meet this threshold could mean that the algorithm failed to steer the synthetic data toward the target distribution at all or simply failed to do so in the allotted number of iterations. For completeness, we provide the average minimum CFD values corresponding to Tables 1 and 2 in Appendix C.1. Additional experimental results are also reported in Appendix C.

## 8 Concluding Remarks

In this work, we presented the *score-difference* (SD) flow as an optimal trajectory for aligning a source distribution with a target distribution. We also showed that this alignment can be performed by working entirely with proxy distributions formed by smoothing the data with additive noise. As a result, while the SD flow defines a deterministic trajectory, its application to noise-injected points adds a stochastic component.

---

[16]See Remark 1 in Appendix B.3.

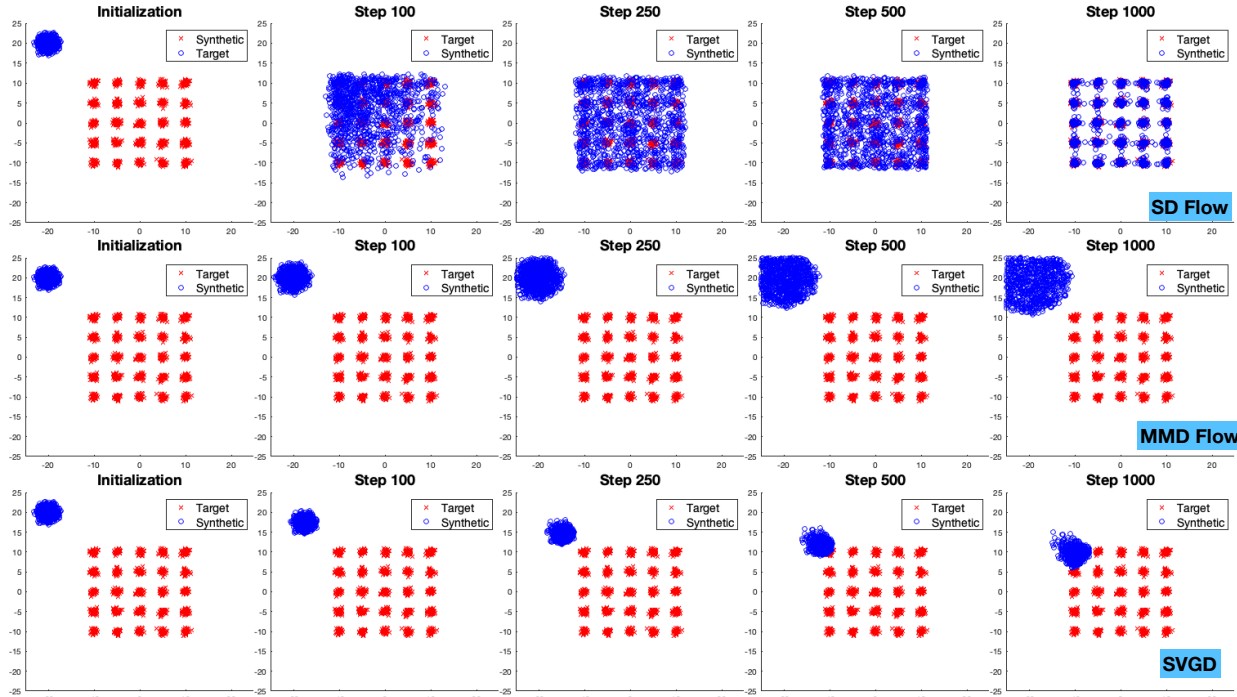

Figure 1: Evolution of synthetic data points from an offset base distribution toward the target distribution of 25 Gaussians over 1000 steps of SD flow (top row), MMD gradient flow (middle row) and SVGD (bottom row) in the no-AdaGrad, full-data, batched, and annealed condition (corresponding to the second row of Table 1). Only SD flow converged in this condition.

SD flow can be used as a direct sampling technique, in which case a sample of base data is converted to a sample from the target distribution via the flow. It can also be used in the training of parametric generative models, in which case generator output is perturbed by the flow to provide the generator with regression targets guaranteed to be closer to the target distribution than the previous output. Unlike most other score-based methods, there are no restrictions on the choice of base or prior distributions. Consistent with this advantage, we have shown that SD flow forms a Schrödinger bridge (Schrödinger, 1932; De Bortoli et al., 2021) between source and target distributions.

We have shown that SD flow emerges in both GANs and diffusion models under certain conditions. The conditions for GANs include the presence of an *optimal* discriminator, which is often unattainable when training with finite, categorically labeled data. By contrast, diffusion models have the comparatively easier task of learning to denoise an image, a task for which the ground truth is more readily represented in the training data. Modern neural denoising architectures that employ attention provide another edge, since they have shown themselves to be extraordinarily capable of capturing patterns in data due to their error-correction and pattern-retrieval characteristics reminiscent of Hopfield networks (Ramsauer et al., 2020).

SD flow supplies a link between IGM methods—namely GANs and diffusion models—that collectively perform well on all three desiderata of the so-called *generative learning trilemma* (Xiao et al., 2021): high sample quality, mode coverage, and fast sampling. Inserting SD flow as an alternative, non-adversarial target-generation module within generator training,[17] replacing the need for an optimal discriminator, could lead to the development of "triple threat" models that produce high-quality, diverse output in a single inference step. The various alternatives we have described in this paper for representing SD flow—namely those based on kernels, denoisers, and density ratios—suggest a variety of opportunities for integrating this approach into a number of IGM frameworks. We look forward to further developments in this direction.

---

[17]See Section 5.1 and Appendix C.4.

Table 2: Convergence probabilities (final three columns) for the SD flow (SD), MMD gradient flow (MMD) and Stein variational gradient descent (SVGD) algorithms after 1000 iterations over five independent trials on the "mystery distribution" particle-optimization problem in $\mathbb{R}^3$ under the experimental conditions described in Section 7.1. Convergence is defined as obtaining a minimum characteristic function distance less than a threshold empirically determined by comparing independent copies of the target distribution (see text). SD flow is the only algorithm to converge under all conditions.

| ADAGRAD | BATCH | CONST | ANNEAL | OFFSET | SD (OURS) | MMD | SVGD |
|---------|-------|-------|--------|--------|-----------|-----|------|
| N | N | N | N | N | 1 | 0 | 0 |
| N | N | N | N | Y | 1 | 0 | 0 |
| N | N | N | Y | N | 1 | 0 | 0 |
| N | N | N | Y | Y | 1 | 0 | 0 |
| N | N | Y | N | N | 1 | 0 | 0 |
| N | N | Y | N | Y | 1 | 0 | 0 |
| N | N | Y | Y | N | 1 | 0 | 0 |
| N | N | Y | Y | Y | 1 | 0 | 0 |
| N | Y | N | N | N | 1 | 0 | 0 |
| N | Y | N | N | Y | 1 | 0 | 0 |
| N | Y | N | Y | N | 1 | 0 | 0 |
| N | Y | N | Y | Y | 1 | 0 | 0 |
| N | Y | Y | N | N | 1 | 0 | 0 |
| N | Y | Y | N | Y | 1 | 0 | 0 |
| N | Y | Y | Y | N | 1 | 0 | 0 |
| N | Y | Y | Y | Y | 1 | 0 | 0 |
| Y | N | N | N | N | 1 | 1 | 1 |
| Y | N | N | N | Y | 1 | 0 | 1 |
| Y | N | N | Y | N | 1 | 1 | 1 |
| Y | N | N | Y | Y | 1 | 0 | 1 |
| Y | N | Y | N | N | 1 | 1 | 1 |
| Y | N | Y | N | Y | 1 | 0 | 1 |
| Y | N | Y | Y | N | 1 | 1 | 1 |
| Y | N | Y | Y | Y | 1 | 0 | 1 |
| Y | Y | N | N | N | 1 | 0 | 1 |
| Y | Y | N | N | Y | 1 | 0 | 1 |
| Y | Y | N | Y | N | 1 | 0 | 1 |
| Y | Y | N | Y | Y | 1 | 0 | 1 |
| Y | Y | Y | N | N | 1 | 0 | 1 |
| Y | Y | Y | N | Y | 1 | 0 | 1 |
| Y | Y | Y | Y | N | 1 | 0 | 1 |
| Y | Y | Y | Y | Y | 1 | 0 | 1 |

**Broader Impact Statement**

Implicit generative modeling in the image and text domains has matured to the point of producing output with an unprecedented level of realism, with other modalities not far behind. It is getting increasingly difficult to tell real data from fake, which is exciting when one reflects on how far the field has come in the past few years but also disturbing when one considers the consequences of advanced IGM methods in the hands of bad actors. We promote the responsible development and use of IGM not only with respect to its deployment but also its training, which should only use data with the proper usage rights secured. We also support continuing research into detecting generated or manipulated data in the effort of counteracting the misuse of this technology and minimizing the societal effects of any misuse.

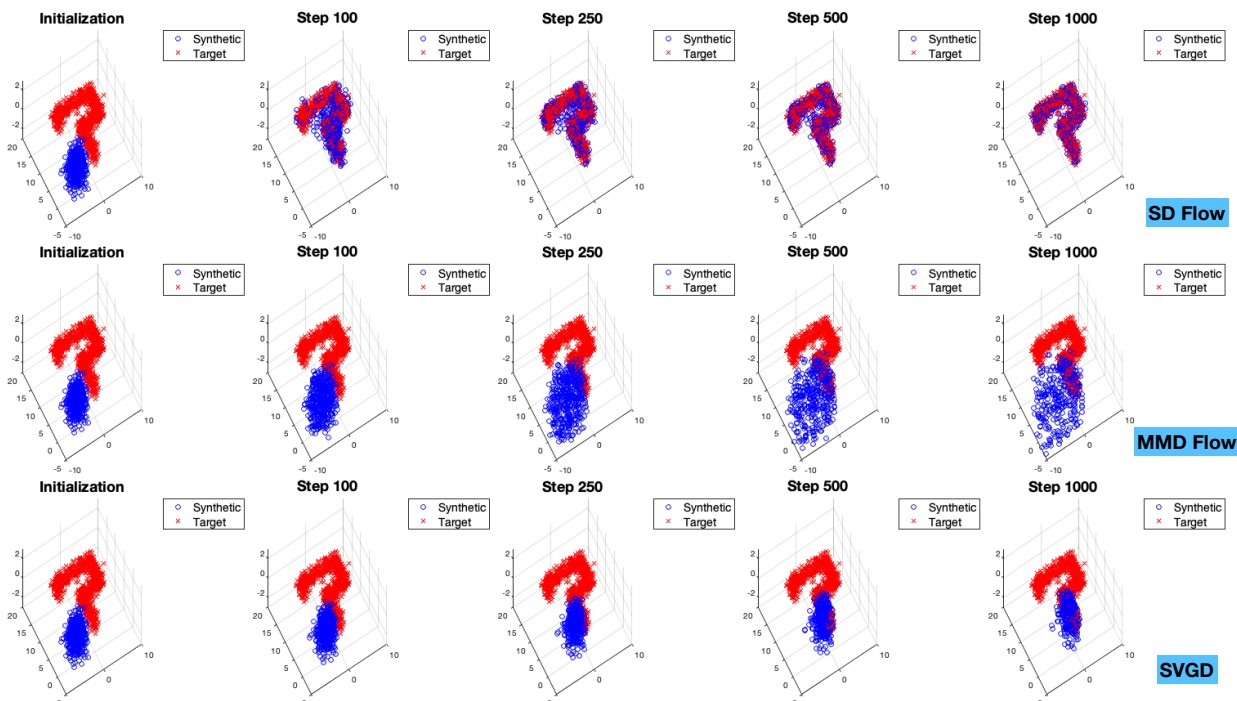

Figure 2: Evolution of synthetic data points from an offset base distribution toward the target distribution of 30 Gaussians over 1000 steps of SD flow (top row), MMD gradient flow (middle row) and SVGD (bottom row) in the no-AdaGrad, full-data, cosine-noise annealed condition (corresponding to the fourth row of Table 2). Only SD flow converged in this condition.

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

## A Guide to the Appendices

In Appendix B.1, we show that the score difference corresponds to the difference between the outputs of *optimal* denoisers corresponding to the target ($p$) and current synthetic ($q_t$) distributions. In Appendix B.2, we describe the evolution of the generative distribution of a GAN under *any* loss. In Appendix B.3, we draw a connection between the kernel definition of SD flow and maximum mean discrepancy (MMD) gradient flow (Arbel et al., 2019). In Appendix B.4, we draw a connection between SD flow and Stein variational gradient descent (SVGD). Additional experimental results are reported in Appendix C.

## B Supplemental Results

### B.1 The Score Difference as the Difference of Optimal Denoisers

We follow an approach similar to that found in Appendix B.3 of Karras et al. (2022) to derive the optimal denoiser for $p$. We define the *denoising loss* by

$$
\begin{aligned}
\mathcal{E}(D_p; \sigma) &= \mathbb{E}_{\boldsymbol{x} \sim p} \mathbb{E}_{\boldsymbol{\epsilon} \sim \mathcal{N}(0, \sigma^2 \boldsymbol{I})} \left[ \| D_p(\boldsymbol{x} + \boldsymbol{\epsilon}; \sigma) - \boldsymbol{x} \|^2 \right] \\
&= \mathbb{E}_{\boldsymbol{x} \sim p} \mathbb{E}_{\boldsymbol{z} \sim \mathcal{N}(\boldsymbol{x}, \sigma^2 \boldsymbol{I})} \left[ \| D_p(\boldsymbol{z}; \sigma) - \boldsymbol{x} \|^2 \right] \\
&= \mathbb{E}_{\boldsymbol{z} \sim \mathcal{N}(\boldsymbol{x}, \sigma^2 \boldsymbol{I})} \mathbb{E}_{\boldsymbol{x} \sim p} \left[ \| D_p(\boldsymbol{z}; \sigma) - \boldsymbol{x} \|^2 \right] \\
&= \int_{\mathbb{R}^d} \underbrace{\mathbb{E}_{\boldsymbol{x} \sim p} \left[ \mathcal{N}(\boldsymbol{z}; \boldsymbol{x}, \sigma^2 \boldsymbol{I}) \| D_p(\boldsymbol{z}; \sigma) - \boldsymbol{x} \|^2 \right]}_{\mathcal{E}(D_p; \boldsymbol{z}, \sigma)} \, \mathrm{d}\boldsymbol{z}.
\end{aligned}
\tag{42}
$$

We can optimize $\mathcal{E}(D_p; \sigma)$ by minimizing the integrand $\mathcal{E}(D_p; \boldsymbol{z}, \sigma)$ pointwise. The optimal denoiser is then given by

$$
D_p^*(\boldsymbol{z}; \sigma) = \arg \min_{D_p(\boldsymbol{z}; \sigma)} \mathcal{E}(D_p; \boldsymbol{z}, \sigma),
\tag{43}
$$

which defines a convex optimization problem. We can then find the global minimum by setting $\nabla_{D_p(\boldsymbol{z}; \sigma)} \mathcal{E}(D_p; \boldsymbol{z}, \sigma)$ to zero, leading to

$$
\begin{aligned}
\mathbb{E}_{\boldsymbol{x} \sim p} \left[ \mathcal{N}(\boldsymbol{z}; \boldsymbol{x}, \sigma^2 \boldsymbol{I}) \nabla_{D_p(\boldsymbol{z}; \sigma)} \| D_p(\boldsymbol{z}; \sigma) - \boldsymbol{x} \|^2 \right] &= \boldsymbol{0} \\
2 D_p(\boldsymbol{z}; \sigma) \mathbb{E}_{\boldsymbol{x} \sim p} \left[ \mathcal{N}(\boldsymbol{z}; \boldsymbol{x}, \sigma^2 \boldsymbol{I}) \right] &= 2 \mathbb{E}_{\boldsymbol{x} \sim p} \left[ \mathcal{N}(\boldsymbol{z}; \boldsymbol{x}, \sigma^2 \boldsymbol{I}) \boldsymbol{x} \right] \\
D_p^*(\boldsymbol{z}; \sigma) &= \frac{\mathbb{E}_{\boldsymbol{x} \sim p} \left[ \mathcal{N}(\boldsymbol{z}; \boldsymbol{x}, \sigma^2 \boldsymbol{I}) \boldsymbol{x} \right]}{\mathbb{E}_{\boldsymbol{x} \sim p} \left[ \mathcal{N}(\boldsymbol{z}; \boldsymbol{x}, \sigma^2 \boldsymbol{I}) \right]},
\end{aligned}
\tag{44}
$$

the final term of which is equal to the first term inside the brackets in equation 18 when $K_\sigma$ is the Gaussian kernel. The derivation for $D_{q_t}^*(\boldsymbol{z}; \sigma)$ is identical, which leads to the result.

### B.2 GAN Dynamics Under General Losses

Since the negative gradient of the GAN loss defines a flow on the generated data $\boldsymbol{y}$, which the generator fits via regression, we can track the evolution of the synthetic distribution $q_t$ within the context of *normalizing flows* (Rezende & Mohamed, 2015; Papamakarios et al., 2021). In particular, the dynamics induced by a

generator loss constitute, in the limit of arbitrarily small steps, a *continuous normalizing flow* whose effect on the synthetic (generated) data distribution is governed by the Liouville equation (see equation 6, Section 2.1),

$$\frac{\partial q_t(\boldsymbol{y}_t)}{\partial t} = \nabla_{\boldsymbol{y}_t} \cdot \left[ q_t(\boldsymbol{y}_t) \nabla_{\boldsymbol{y}} \mathcal{L} \right], \tag{45}$$

a continuity equation with solution

$$q_t(\boldsymbol{y}_t) = q_0(\boldsymbol{y}_0) \exp\left( \int_0^t \mathrm{Tr}\left[ \boldsymbol{H}_{\mathcal{L}}(\boldsymbol{y}_\tau) \right] \mathrm{d}\tau \right) = q_0(\boldsymbol{y}_0) \exp\left( \int_0^t \nabla_{\boldsymbol{y}_\tau}^2 \mathcal{L} \, \mathrm{d}\tau \right), \tag{46}$$

where $\boldsymbol{H}_{\mathcal{L}}(\boldsymbol{y}_\tau)$ is the Hessian matrix of the loss $\mathcal{L}$ evaluated at $\boldsymbol{y}_\tau$, whose trace is the Laplacian $\nabla_{\boldsymbol{y}}^2 \mathcal{L} = \sum_i \partial^2/\partial y_i^2 \mathcal{L}$. Taking the logarithm of both sides of equation 46 yields the solution to the *instantaneous change of variables* differential equation (Chen et al., 2018; Grathwohl et al., 2018). Note that this evolution holds for any generator loss $\mathcal{L}$ and does not make any assumptions about an optimal discriminator.

## B.3 Relation to MMD Gradient Flow

In the study of reproducing kernel Hilbert spaces (RKHS), the Gaussian kernel $K_\sigma(\boldsymbol{z}, \boldsymbol{x})$ is known as a *characteristic* kernel (Sriperumbudur et al., 2011). This means that the mapping $\varphi_p(\boldsymbol{z}) = \mathbb{E}_{\boldsymbol{x} \sim p}[K_\sigma(\boldsymbol{z}, \boldsymbol{x})]$ is *injective*, and $\varphi_p(\boldsymbol{z}) = \varphi_q(\boldsymbol{z})$ for all $\boldsymbol{z}$ if and only if $p = q$. This forms the basis of the *maximum mean discrepancy* (MMD), which is equal to the Hilbert space norm $\|\mathcal{W}_{p,q}\|_{\mathcal{H}}$, where

$$\mathcal{W}_{p,q}(\boldsymbol{z}) = \varphi_q(\boldsymbol{z}) - \varphi_p(\boldsymbol{z}) = \mathbb{E}_{\boldsymbol{y} \sim q}[K_\sigma(\boldsymbol{z}, \boldsymbol{y})] - \mathbb{E}_{\boldsymbol{x} \sim p}[K_\sigma(\boldsymbol{z}, \boldsymbol{x})] \tag{47}$$

is known as the *witness function* (Gretton et al., 2012; Arbel et al., 2019).

In the theory of optimal transport, we wish to efficiently transport "mass" from an initial distribution $q_0$ to a target distribution $p$, which we can do by defining a flow from $q_0$ to $p$ via intermediate distributions $q_t$. One such flow is defined by the solution to

$$\frac{\partial q_t}{\partial t} = \nabla \cdot \left[ q_t \nabla \mathcal{W}_{p,q_t} \right], \tag{48}$$

another instance of the Liouville equation (6, 45) that defines a McKean-Vlasov process (McKean Jr, 1966) with dynamics

$$\begin{aligned} \mathrm{d}\boldsymbol{z}_t &= -\nabla_{\boldsymbol{z}_t} \mathcal{W}_{p,q_t}(\boldsymbol{z}_t) \, \mathrm{d}t \\ &= (\mathbb{E}_{\boldsymbol{x} \sim p}[\nabla_{\boldsymbol{z}_t} K_\sigma(\boldsymbol{z}_t, \boldsymbol{x})] - \mathbb{E}_{\boldsymbol{y} \sim q}[\nabla_{\boldsymbol{z}_t} K_\sigma(\boldsymbol{z}_t, \boldsymbol{y})]) \, \mathrm{d}t, \end{aligned} \tag{49}$$

where $\boldsymbol{z}_0 \sim q_0$. The results of Section 3.1 suggest that, in the limit of infinite data, this direction is proportional to $\nabla_{\boldsymbol{z}_t} p(\boldsymbol{z}_t; \sigma) - \nabla_{\boldsymbol{z}_t} q(\boldsymbol{z}_t; \sigma)$.

For the Gaussian kernel, we have

$$\nabla_{\boldsymbol{z}_t} K_\sigma(\boldsymbol{z}_t, \boldsymbol{x}) = K_\sigma(\boldsymbol{z}_t, \boldsymbol{x}) \left( \frac{\boldsymbol{x} - \boldsymbol{z}_t}{\sigma^2} \right),$$

and for discrete time and finite data we can write equation 49 as

$$\Delta \boldsymbol{z}_t = \frac{1}{N} \sum_{i=1}^{N} \left[ K_\sigma(\boldsymbol{z}_t, \boldsymbol{x}_i) \left( \frac{\boldsymbol{x}_i - \boldsymbol{z}_t}{\sigma^2} \right) \right] - \frac{1}{M} \sum_{j=1}^{M} \left[ K_\sigma(\boldsymbol{z}_t, \boldsymbol{y}_j) \left( \frac{\boldsymbol{y}_j - \boldsymbol{z}_t}{\sigma^2} \right) \right] \tag{50}$$

$$= \sum_{i=1}^{N} w_i^{(p)} \boldsymbol{x}_i - \sum_{j=1}^{M} w_j^{(q_t)} \boldsymbol{y}_j + \left( \sum_{j=1}^{M} w_j^{(q_t)} - \sum_{i=1}^{N} w_i^{(p)} \right) \boldsymbol{z}_t, \tag{51}$$

where $w_i^{(p)} = K_\sigma(\boldsymbol{z}_t, \boldsymbol{x}_i)/(N\sigma^2)$ and $w_j^{(q_t)} = K_\sigma(\boldsymbol{z}_t, \boldsymbol{y}_j)/(M\sigma^2)$. This process defines the *MMD gradient flow* (Arbel et al., 2019). The kernel version of SD flow (equation 18) can also be written in the form

of equation 51 by setting $w_i^{(p)} = \frac{1}{2} K_\sigma(\boldsymbol{z}_t, \boldsymbol{x}_i)/\sum_{i=1}^N K_\sigma(\boldsymbol{z}_t, \boldsymbol{x}_i)$ and $w_j^{(q_t)} = \frac{1}{2} K_\sigma(\boldsymbol{z}_t, \boldsymbol{x}_i)/\sum_{j=1}^M K_\sigma(\boldsymbol{z}_t, \boldsymbol{y}_j)$, which causes the $\boldsymbol{z}_t$ term to vanish.

There are practical consequences of this difference in weighting schemes between methods, which put the MMD gradient flow at a disadvantage in some conditions, as discussed in the following remark.

**Remark 1** *If $p$ and $q_t$ are far apart, for a point $\boldsymbol{z}_t = \boldsymbol{y} + \sigma\boldsymbol{\epsilon}$, with $\boldsymbol{y} \sim q_t$ and a small kernel bandwidth $\sigma$, equation 51 shows that under the MMD framework $\sum_j w_j^{(q_t)} \approx 1/(M\sigma^2)$ and $\sum_i w_i^{(p)} \approx 0$. This suggests that under these conditions, the MMD gradient flow direction $\Delta\boldsymbol{z}_t^{MMD}$ will be nearly parallel to $\boldsymbol{z}_t - \boldsymbol{y} = \sigma\boldsymbol{\epsilon}$, which would have the effect of increasing the variance of $q_t$ while not necessarily pushing it toward $p$. Evidence of this variance-exploding effect emerged in our experiments and also apparent in the authors' original paper (e.g. Appendix G.2 therein). Normalizing the weights in equation 51 to sum to one resolves this issue, however, and MMD gradient flow and SD flow then become equivalent.*

**Remark 2** *Our method prescribes that we inject noise at a level equal to the kernel bandwidth in order to sample from the noise-smoothed proxy distribution. In contrast, with MMD gradient flow the noise level is a separate parameter that essentially controls a regularization effect. In that case, this added noise is typically at a level far greater than that of the kernel bandwidth, which remains fixed during training.*

### B.4 Relation to Stein Variational Gradient Descent (SVGD)

Another statistical distance that measures the discrepancy between distributions $p$ and $q$ is the *Stein discrepancy* (Gorham & Mackey, 2015),

$$\mathbb{D}_S(p\|q) = \mathbb{E}_{\boldsymbol{x}\sim q}[\mathrm{Tr}\left(\mathcal{A}_p \mathbf{f}(\boldsymbol{x})\right)], \tag{52}$$

where $\mathcal{A}_p$ is the Stein operator defined in equation 9, and $\mathbf{f}$ is a function that vanishes on the boundary of the support of $p, q$ or (equivalently) behaves such that $\mathbf{f}(\boldsymbol{x})p(\boldsymbol{x}) \to \mathbf{0}$ and $\mathbf{f}(\boldsymbol{x})q(\boldsymbol{x}) \to \mathbf{0}$ as $\|\boldsymbol{x}\| \to \infty$.

The discrepancy 52 vanishes if and only if $p = q$, which can be seen by expanding out equation 52 as in equation 10,

$$\mathbb{E}_{\boldsymbol{x}\sim q}[\mathrm{Tr}\left(\mathcal{A}_p \mathbf{f}(\boldsymbol{x})\right)] = \mathbb{E}_{\boldsymbol{x}\sim q}\left[\left(\nabla_{\boldsymbol{x}}\log p(\boldsymbol{x}) - \nabla_{\boldsymbol{x}}\log q(\boldsymbol{x})\right)^\top \mathbf{f}(\boldsymbol{x})\right], \tag{53}$$

since for nontrivial functions $\mathbf{f}$, the above vanishes only when $\nabla_{\boldsymbol{x}}\log p(\boldsymbol{x}) = \nabla_{\boldsymbol{x}}\log q(\boldsymbol{x})$. When $\mathbf{f}$ is restricted to bounded functions within a reproducing kernel Hilbert space (RKHS), one obtains the *kernel Stein discrepancy* (Liu et al., 2016).

As reported in Section 2.2, Liu & Wang (2016) establish a link between the kernel Stein discrepancy and the variation in the KL divergence, which leads to their derivation of Stein variational gradient descent (SVGD) as an optimal direction for reducing the KL divergence between $q$ and $p$ when operating in a RKHS:

$$\begin{aligned} \phi(\boldsymbol{z}) &= \mathbb{E}_{\boldsymbol{x}\sim q}\left[\nabla_{\boldsymbol{x}}\log p(\boldsymbol{x})K(\boldsymbol{z}, \boldsymbol{x}) + \nabla_{\boldsymbol{x}}K(\boldsymbol{z}, \boldsymbol{x})\right] \\ &= \mathbb{E}_{\boldsymbol{x}\sim q}\left[\left(\nabla_{\boldsymbol{x}}\log p(\boldsymbol{x}) - \nabla_{\boldsymbol{x}}\log q(\boldsymbol{x})\right)K(\boldsymbol{z}, \boldsymbol{x})\right], \end{aligned} \tag{54}$$

which shows SVGD to be the kernel-weighted average of the score difference. Ba et al. (2021) provide a separate analysis of the connection between SVGD and MMD gradient flow.

## C  Additional Experimental Results

### C.1  Average CFD Values for Particle-Optimization Experiments

In Sections 7.2.1 and 7.2.2, we reported convergence probabilities for SD flow, MMD gradient flow, and SVGD under various experimental conditions. Convergence was defined as achieving a minimum characteristic function distance (CFD) below a threshold empirically determined by multiple comparisons of independent copies of the target distribution (0.0651 for the 25-Gaussian grid problem in $\mathbb{R}^2$ and 0.0612 for the 30-Gaussian "mystery distribution" in $\mathbb{R}^3$). In Tables 3 and 4, we report average CFD minimum values achieved by the algorithms after 1000 steps over five separate trials.

Table 3: Average minimum CFD (final three columns) for the SD flow (SD), MMD gradient flow (MMD) and Stein variational gradient descent (SVGD) algorithms after 1000 iterations over five independent trials on the 25-Gaussian particle-optimization problem in $\mathbb{R}^2$ under the experimental conditions described in Section 7.1.

| ADAGRAD | BATCH | CONST | ANNEAL | SD (OURS) | MMD | SVGD |
|---------|-------|-------|--------|-----------|-----|------|
| N | N | N | N | 0.0007 | 0.6129 | 0.8495 |
| N | N | N | Y | 0.0007 | 0.6111 | 0.8521 |
| N | N | Y | N | 0.0007 | 0.5816 | 0.7878 |
| N | N | Y | Y | 0.0009 | 0.5837 | 0.7978 |
| N | Y | N | N | 0.0374 | 0.9382 | 0.7944 |
| N | Y | N | Y | 0.0357 | 0.9365 | 0.7863 |
| N | Y | Y | N | 0.0371 | 0.9230 | 0.7454 |
| N | Y | Y | Y | 0.0389 | 0.9249 | 0.7494 |
| Y | N | N | N | 0.0006 | 0.1584 | 0.0033 |
| Y | N | N | Y | 0.0006 | 0.1622 | 0.0031 |
| Y | N | Y | N | 0.0006 | 0.1566 | 0.0029 |
| Y | N | Y | Y | 0.0006 | 0.1553 | 0.0028 |
| Y | Y | N | N | 0.0359 | 0.1884 | 0.0412 |
| Y | Y | N | Y | 0.0399 | 0.1895 | 0.0414 |
| Y | Y | Y | N | 0.0381 | 0.1892 | 0.0377 |
| Y | Y | Y | Y | 0.0385 | 0.1902 | 0.0359 |

### C.2 Data-Set Interpolation

Standard score-based generative modeling is designed such that the end of the forward process is a Gaussian distribution. While this has the advantage of defining a prior that is easy to sample from for the reverse, generative process, it limits the flexibility of the method. Recent work on approximating a *Schrödinger bridge* between source and target distributions (De Bortoli et al., 2021) relaxes this limitation, but the method itself is relatively complicated.

SD flow, on the other hand, is (in our opinion) a simpler and more intuitive method that also solves the Schrödinger bridge problem (see Section 2.3) and places no restrictions on the distributions $p$ and $q$. It is therefore also capable of performing interpolation between arbitrary data sets. The results of one such interpolation experiment are shown in Figure 3. The figure actually shows *two* interpolation experiments: The first evolves 1024 points of the "Swiss roll" data toward the "mystery" distribution (Section 7.2.2) in $\mathbb{R}^3$, while the second evolves from the "mystery" distribution to the "Swiss roll." The same cosine variance schedule as in Section 7.2.2 was employed.

### C.3 Nearest-Neighbor Analysis

It is important to note that SD flow does not cause the synthetic data to collapse to nearest neighbors in the target distribution. In Figure 4, we show the distribution of distances from points in the synthetic distribution to their first nearest neighbors in the target distribution (shaded in green) for the particle-optimization experiment described in Section 7.2.2. Note the overlap with the distribution of distances between target data points and their first nearest neighbors (excluding themselves) in the target distribution. Overfitting to the target data would result in a large concentration of mass near zero for the synthetic data.

### C.4 Model Optimization

Although we do not run experiments on high-dimensional image data in the present work for the reasons described in Section 7, we report here an experiment using the model-optimization application (Algorithm 2) on "high"-dimensional data in $\mathbb{R}^{50}$. Here the scare quotes acknowledge that this dimensionality is far lower

Table 4: Average minimum CFD (final three columns) for the SD flow (SD), MMD gradient flow (MMD) and Stein variational gradient descent (SVGD) algorithms after 1000 iterations over five independent trials on the "mystery distribution" particle-optimization problem in $\mathbb{R}^3$ under the experimental conditions described in Section 7.1.

| ADAGRAD | BATCH | CONST | ANNEAL | OFFSET | SD (OURS) | MMD | SVGD |
|---|---|---|---|---|---|---|---|
| N | N | N | N | N | 0.0002 | 0.1020 | 0.2407 |
| N | N | N | N | Y | 0.0026 | 0.7050 | 0.6994 |
| N | N | N | Y | N | 0.0004 | 0.1020 | 0.2407 |
| N | N | N | Y | Y | 0.0020 | 0.7050 | 0.6995 |
| N | N | Y | N | N | 0.0005 | 0.1186 | 0.2761 |
| N | N | Y | N | Y | 0.0006 | 0.5953 | 0.7006 |
| N | N | Y | Y | N | 0.0005 | 0.1185 | 0.2762 |
| N | N | Y | Y | Y | 0.0006 | 0.5953 | 0.7006 |
| N | Y | N | N | N | 0.0146 | 0.1689 | 0.1804 |
| N | Y | N | N | Y | 0.0136 | 0.8250 | 0.6060 |
| N | Y | N | Y | N | 0.0157 | 0.1734 | 0.1725 |
| N | Y | N | Y | Y | 0.0172 | 0.8231 | 0.6051 |
| N | Y | Y | N | N | 0.0159 | 0.2086 | 0.2410 |
| N | Y | Y | N | Y | 0.0157 | 0.8348 | 0.6168 |
| N | Y | Y | Y | N | 0.0167 | 0.2045 | 0.2359 |
| N | Y | Y | Y | Y | 0.0168 | 0.8404 | 0.6265 |
| Y | N | N | N | N | 0.0039 | 0.0029 | 0.0059 |
| Y | N | N | N | Y | 0.0038 | 0.1945 | 0.0057 |
| Y | N | N | Y | N | 0.0037 | 0.0033 | 0.0059 |
| Y | N | N | Y | Y | 0.0042 | 0.1938 | 0.0060 |
| Y | N | Y | N | N | 0.0056 | 0.0039 | 0.0077 |
| Y | N | Y | N | Y | 0.0056 | 0.0847 | 0.0081 |
| Y | N | Y | Y | N | 0.0056 | 0.0041 | 0.0082 |
| Y | N | Y | Y | Y | 0.0056 | 0.0841 | 0.0083 |
| Y | Y | N | N | N | 0.0154 | 0.0972 | 0.0175 |
| Y | Y | N | N | Y | 0.0163 | 0.3141 | 0.0185 |
| Y | Y | N | Y | N | 0.0148 | 0.0972 | 0.0176 |
| Y | Y | N | Y | Y | 0.0162 | 0.3117 | 0.0164 |
| Y | Y | Y | N | N | 0.0146 | 0.1122 | 0.0242 |
| Y | Y | Y | N | Y | 0.0162 | 0.2915 | 0.0228 |
| Y | Y | Y | Y | N | 0.0165 | 0.1111 | 0.0222 |
| Y | Y | Y | Y | Y | 0.0149 | 0.2922 | 0.0238 |

than the thousands to millions of dimensions typical in high-resolution image data, but it is high enough to exhibit the problematic, intuition-challenging characteristics of high-dimensional data in general.

Specifically, it is well known that as data dimensionality grows, the ratio of the distance of a point to its *farthest* neighbor, $D_{\max}$, and the distance to its *nearest* neighbor, $D_{\min}$, tends toward unity. The ratio $D_{\max}/D_{\min}$ drops precipitously in lower dimensions before leveling off at around 30 dimensions and very slowly approaching an asymptote of one afterward (Beyer et al., 1999). We therefore chose $\mathbb{R}^{50}$ as a reasonable setting to challenge our approach in high dimensions.

We generated a ground-truth target distribution by randomly populating a $50 \times 25$ matrix $\boldsymbol{B}$ with values drawn from $\mathcal{N}(0, 0.25\boldsymbol{I})$ and a 50-vector $\boldsymbol{\mu}$ with values drawn from $\mathcal{N}(10, \boldsymbol{I})$. Target data samples from $\mathcal{N}(\boldsymbol{\mu}, \boldsymbol{B}\boldsymbol{B}^\top)$ were then generated[18] by drawing samples $\boldsymbol{\xi} \sim \mathcal{N}(\boldsymbol{0}, \boldsymbol{I}) \in \mathbb{R}^{25}$ and forming $\boldsymbol{x} = \boldsymbol{B}\boldsymbol{\xi} + \boldsymbol{\mu}$. The

---

[18]Technically, this is not completely well defined as a normal distribution, since $\boldsymbol{B}\boldsymbol{B}^\top$ is not of full rank.

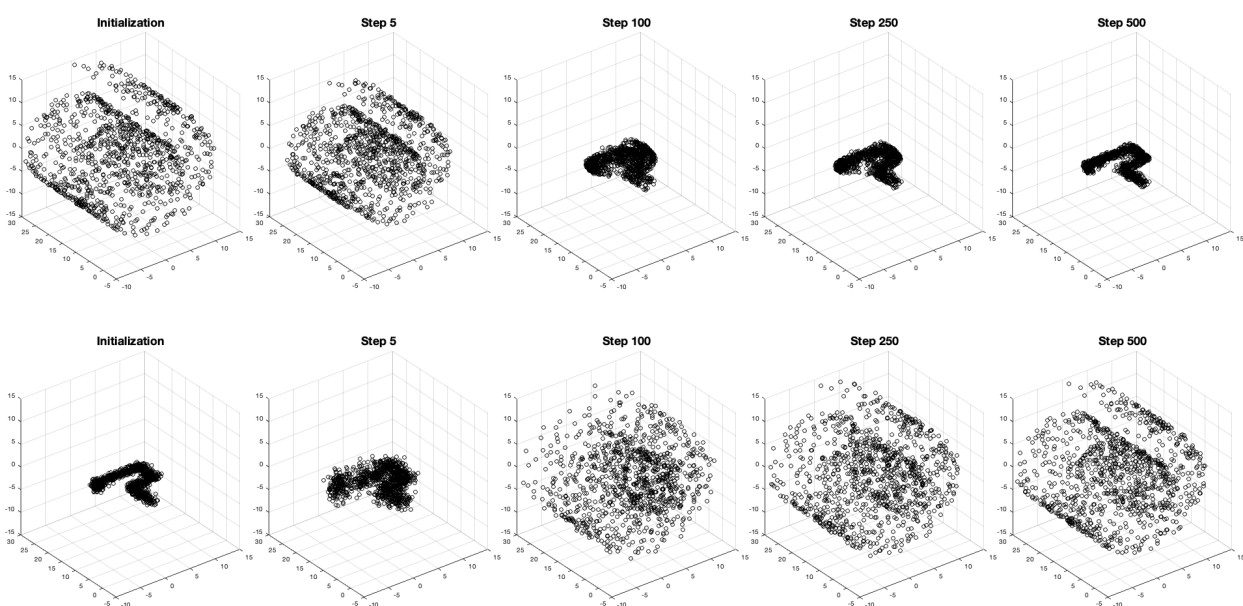

Figure 3: Top: Data-set interpolation via evolution of 1024 points from the "Swiss roll" distribution to the "mystery" distribution in $\mathbb{R}^3$. Bottom: The reverse interpolation, from the "mystery" distribution to the "Swiss roll" distribution.

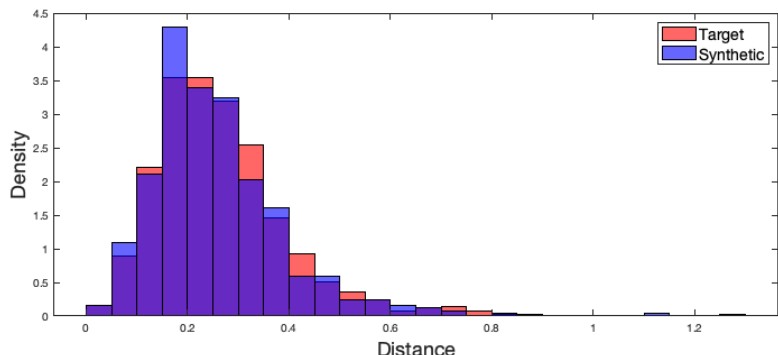

Figure 4: Distribution of distances from synthetic (blue) and target (red) data points to their first nearest neighbors in the target distribution.

model parameters to be learned were a $50 \times 25$ matrix $\hat{\boldsymbol{B}}$, initialized from $\mathcal{N}(0, 0.01\boldsymbol{I})$, and a 50-vector $\hat{\boldsymbol{\mu}}$, initialized to all zeros. This model can be interpreted as a single-layer linear neural network, but it is most important to note that it exactly matches the capacity of the data-generating model.

If we retained the input vectors $\boldsymbol{\xi} \in \boldsymbol{\Xi}$ for the outputs $\boldsymbol{x} \in \boldsymbol{X}$, then the task of learning the parameters would be fairly straightforward in the context of a regression problem on paired data $\{(\boldsymbol{\xi}, \boldsymbol{x})\}$. But in the general IGM problem, we have only *unpaired* data to work with, so we assume that all information about the target data inputs is unavailable.

We performed 1000 steps of SD flow using Algorithm 2 with a *constant* noise schedule of 10 times the average distance of the initial synthetic (base) distribution to first nearest neighbors in the target distribution

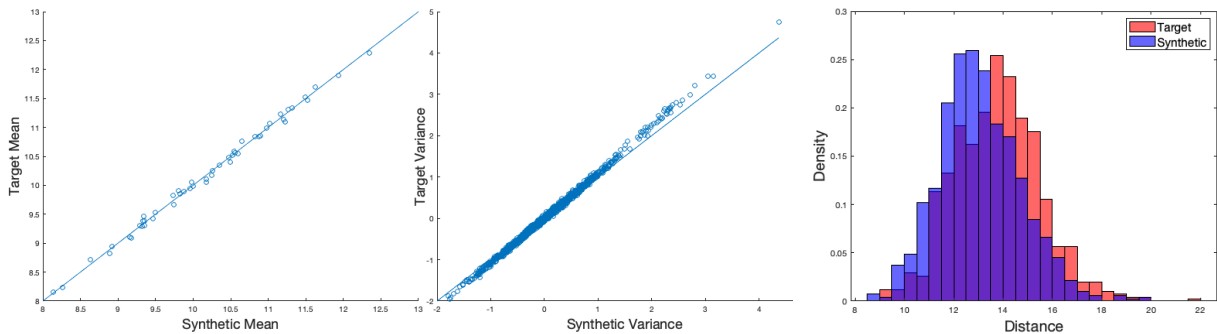

Figure 5: Model optimization results in $\mathbb{R}^{50}$ using a constant noise schedule. SD flow allows a parametric model to be learned that very closely matches the target mean ($\boldsymbol{\mu}$ versus $\hat{\boldsymbol{\mu}}$, left panel) and the elements of the covariance matrix ($\boldsymbol{B}\boldsymbol{B}^{\top}$ vs $\hat{\boldsymbol{B}}\hat{\boldsymbol{B}}^{\top}$, center panel). Diagonals are included for reference. Nearest-neighbor analysis showed no overfitting of the data (right panel) but showed a slightly lower average distance to nearest neighbors in the target set than exhibited by the target data relative to itself.

(corresponding to $\sigma^2 > 700$),[19] with a batch size of 1024, an SD flow step size of $\eta = 1$, and a regression learning rate of $\lambda = 10^{-3}$. Other than brief experimentation to set reasonable values, no effort was made to optimize these hyperparameters.

The results of this experiment are shown in Figure 5. Despite (or perhaps *because of*) a massive and constant injection of noise, SD flow successfully fit the target distribution. Analysis of nearest neighbors once again showed that SD flow did not overfit to the target distribution, although there was a very slight shift toward lower distances between synthetic data and their nearest neighbors in the target distribution as compared with the target data's nearest-neighbor distances relative to itself.

This experiment provides a basic proof of concept for a much more general procedure, one that can benefit from more sophisticated model architectures specifically suited to the problem at hand. For instance, for image generation, the kernel-based specification of SD flow from Section 3.1 can be exchanged for the denoising-based specification from Section 3.2, allowing one to take advantage of attention-equipped U-net denoising architectures. Furthermore, methods can be mixed and matched: A denoising model can be employed for the $p$ score component while a kernel-based estimate can be used for the $q_t$ score component, for example.

---

[19]We found that using a higher amount of noise somewhat improved the convergence profile of the algorithm.

