# OpenReview forum: "The Score-Difference Flow for Implicit Generative Modeling"
_TMLR — Accepted by TMLR_

### Review · Reviewer_SSdk · 2023-04-12

**Summary Of Contributions:**

In this paper, the authors introduce Score Difference, a new method to solve generative modeling problems. The score difference can be found as the direction of maximal descent in a forward KL flow. However, since the target distribution $p$ is not known and only accessible through samples, the authors define a mollified version of the target distribution by adding noise to it. Doing so and leveraging Tweedie's identity, they are able to write the score difference as a difference of ratio of expectation which are tractable. The authors then show the links between their approach and two popular generative models: GANs and diffusion models. Toy experiments are reported in the appendix.

**Audience:**

Yes

**Claims And Evidence:**

No

**Requested Changes:**

* The authors should address the main weaknesses highlighted in the previous section.
* In particular I think that the paper could benefit from a deeper analysis of the link with diffusion models and SVGD.
* A better experimental setting (image or other higher dimensional tasks with a comparison with baselines) is necessary to assess the experimental properties of SD Flows.

**Strengths And Weaknesses:**

STRENGTHS:
* I think the link with diffusion models is interesting. There is potential here to build an interesting connection between continuous normalizing flows, the probability flow interpretation of diffusion models and score difference.

WEAKNESSES:

* The striking weakness for me is the lack of experiments. This paper deals with "Generative Modeling" and so should at least showcase how the interpretation of diffusion models and GANs in terms of Score Difference yield algorithms with better properties or at least provide a better understanding of their theoretical properties. I don't really see what new insights or properties of diffusion models/GANs can be inferred from their reformulation as Score Differences. If this paper is a theoretical paper I think that these contributions should be made clearer and that a proper theorem should be written about them. If the paper is a methodological/experimental paper then the method should be evaluated properly and compared with other methods for generative modeling and dataset interpolation. As of now, the only experiments are toy (3d up to 50d) with no baseline comparison.

* The paper is not very well written. It is hard to decipher what are the contributions of the authors (see first point). For example, what is the main goal of Section 4? We start with a general decomposition of a loss in Section 4.1 and then move on to continuous normalizing flows in Section 4.2 to finish with the link with SD Flows in Section 4.3. There is no real connections between these sections. For example, what purpose serves Section 4.2?

* I think the authors should do a better job at discussing the links between their approach and existing approaches. For example,  [1] explores another type of annealing (not the same as the mollified approach considered in this work). Overall, I think that a review of existing SVGD generalizations and how they relate to the current approach could be useful. Also the authors do not discuss the probability flow approach of diffusion models [2]. I think that this is more related to the current approach as there is no noise injected at each step in that case.

* Regarding the data interpolation task, I think the authors should discuss and compare their approach with [3,4,5]. [3,4] are Schrodinger Bridge approaches. The authors comment in the main text that "SD flow is capable of performing data-set interpolation in a manner similar to Diffusion Schrodinger Bridges while being more straightforward to apply". However SD Flow does not solve the Schrodinger Bridge problem. In particular, one of the main point of [3,4] is that the obtained flow is an Optimal Transport map. I also want to point out that the idea of "get from point A to point B as quickly as possible, we want to move not only toward B but also away from A" is already present in Schrodinger Bridge approaches. Indeed if we consider a first forward process given by a Brownian motion going from distribution A to a noised version of A, $(X_t^0) = (B_t)$, then the backward process starting from distribution B is given $(X_t^1)$ satisfying $\mathrm{d} X_t^1 = \nabla \log p_{1-t}^0(X_t^1) \mathrm{d} t + \mathrm{d} B_t$, where $p_t^0$ is the density of $X_t^0$. Then, the backward of this backward dynamics initialized at distribution A is called $(X_t^2)$ and satisfies $\mathrm{d} X_t^2 = (-\nabla \log p_{t}^0(X_t^2) +\nabla \log p_{1-t}^1(X_t^2)) \mathrm{d} t + \mathrm{d} B_t$. Hence, we recover a score difference there too. Note that Schrodinger Bridge approaches go one step further by accumulating the scores (which are approximated by neural networks). If one is not interested in the Optimal Transport properties of the flow then [5] is a very efficient approach based on Diffusion models for Image to Image translation.

* I found some of the claims to be not backed-up by experiments or theory. In particular, in the discussion the authors claim that their approach "perform well on all three desiderata of the so-called generative learning trilemma -- high sample quality, mode coverage, fast sampling. High sample quality is not demonstrated in this work since no high dimensional experiment is proposed. Mode coverage is relatively illustrated in low dimensional settings with the coverage of a mixture of Gaussian distributions. Finally it is not clear at all that fast sampling is a property of SD flow. It would seem to suffer from the same drawbacks of diffusion models, i.e. multiple iteratives steps at inference times. While distillation strategies could be used it is not presented in the current paper.

[1] D'Angelo, Fortuin (2021) -- Annealed Stein Variational Gradient Descent

[2] Song, Durkan, Murray, Ermon (2021) -- Maximum Likelihood Training of Score-Based Diffusion Models

[3] De Bortoli, Thornton, Heng, Doucet (2021) -- Diffusion Schrödinger Bridge with Applications to Score-Based Generative Modeling

[4] Chen, Liu, Theodorou (2021) -- Likelihood Training of Schrödinger Bridge using Forward-Backward SDEs Theory

[5] Su, Song, Meng, Ermon (2022) -- Dual Diffusion Implicit Bridges for Image-to-Image Translation

---

> ### Author Response · Authors · 2023-04-13
> **Initial Response to Reviewer SSdk**
>
> We thank the reviewer for the time invested in preparing the review.  While a thorough, point-by-point response will follow, we felt it important to craft an initial response to address several broader points.  Specifically, (1) we wish to make it clear how we view the paper and its contributions, and (2) we wish to correct a misapprehension the reviewer has about one aspect of what the paper claims.
>
> This is a theory paper, which is why we relegated any experimental results to the appendix.  We do not present our work in the theorem-proof format, opting instead for a presentation that we hoped the reader would find more intuitive, insightful, and engaging.  We can certainly revisit our choice of presentation if it is not having the intended effect.  However, we believe that all of our claims are given with sufficient mathematical justification.
>
> We tried to be as clear as possible about the theoretical contributions around which the paper is organized, which we enumerate at the end of the paper's introduction and paraphrase here:
> * In Section 2, we derive the SD flow and show that it optimally reduces the KL divergence. (The optimality with respect to the KL divergence can be seen as a second derivation of SD flow, independent of the derivation from probability flow.)  Despite the large and growing literature on probability flow, we are unaware of any literature that focuses on the score difference as a flow or perturbation direction for data.
> * In Section 3, we describe the alignment of proxy distributions, which correspond to the source and target distributions corrupted by Gaussian noise.  Working with such modified distributions is not in itself a new idea (see [1], for example).  However, demonstrating that operating *exclusively* on these proxies is sufficient to align the original distributions provides, we believe, useful perspective for the sections that follow.
> * In Section 4, we show that GAN generator training implicitly induces a flow on the generated data determined by the loss and that for certain choices of loss, that flow is the SD flow when the discriminator is optimal.  We are unaware of anywhere else in the literature where this perspective is presented.  To the reviewer's point, while we find the discussion in Section 4.2 of the evolution of the generative distribution under *any* loss useful and interesting, it is not essential to the overall message of Section 4, and we would be happy to move it or cut it altogether.  The overall message, however, relates to SD flow's presence in GANs under optimal conditions.  This is important, since GANs are known to tackle two out of the three challenges of the generative modeling trilemma, namely high-quality samples and fast sampling.
> * In Section 5, we show that updating proxy distributions (Section 3) under SD flow is equivalent to denoising diffusion under certain conditions.  This is important, since diffusion models are known to tackle two out of the three challenges of the generative modeling trilemma, namely high-quality samples and mode coverage.
>
> The reviewer has unfortunately misread our commentary on the generative modeling trilemma.  We do not claim to have shown in the paper that we have directly tackled all three challenges.  Rather, in the line the reviewer partially quotes, we write: "SD flow supplies a link between IGM methods [namely GANs and diffusion models] that collectively perform well on all three desiderata of the so-called *generative learning trilemma*. ... "  (We have inserted brackets to clarify what we are referring to.)  The idea we wish to suggest is that by inserting SD flow as a target-generation module in generator training (removing the need for an optimal discriminator), the resulting generator should be able to perform well on all three challenges.  This is a proposal that is motivated by the theory developed in the paper.
>  We will make every effort to ensure that this point is clearer in any revisions.
>
> Some of the areas of additional discussion the reviewer suggests are interesting, especially with regard to SVGD and Schrödinger bridge (SB) approaches.  (We will address the reviewer's comments on the SB problem in subsequent remarks.)  However, we once again emphasize that as a theory paper, extensive experimentation on high-dimensional image data falls outside the scope of the paper.  We also note that the experiments that we do present are on target and source data no less complex than what is presented in [2], which the reviewer cites in the original review.
>
> We present these initial remarks in the hope that the reviewer will revisit our central claims and reconsider whether they are supported by sound mathematical arguments, which we believe they are.
>
> [1] Zhang, M., et al. "Spread divergence." International Conference on Machine Learning. PMLR, 2020.
> [2] D'Angelo, F., and Fortuin, V. "Annealed stein variational gradient descent." arXiv preprint arXiv:2101.09815 (2021).

---

> > ### Author Response · Authors · 2023-04-15
> > **SD Flow and the Schrödinger Bridge Problem**
> >
> > In the submitted version of our manuscript, we wrote: "SD flow is capable of performing data-set interpolation in a manner similar to diffusion Schrödinger bridges," which Reviewer SSdk quotes in the posted review.  Our intention here was to call out that, unlike diffusion models but like Schrödinger bridges, SD flow can interpolate between *arbitrary* distributions, with no restrictions placed on the prior.  We did not set out to specifically address the Schrödinger bridge (SB) problem.  However, Reviewer SSdk's remarks inspired us to examine SD flow in the context of the SB problem, and we discovered that SD flow does appear to provide a solution to this problem.
> >
> > The SB problem considers the following forward and backward SDEs [1, 2, 3]: $$dx_t = [\mu(x_t,t) + \gamma(t) \nabla \log \Psi(x_t, t)] dt + \sqrt{\gamma(t)} d\omega$$ and $$dx_t = [\mu(x_t,t) - \gamma(t) \nabla \log \hat{\Psi}(x_t, t)] dt + \sqrt{\gamma(t)} d\hat{\omega},$$ where $\Psi$ and $\hat{\Psi}$ are potentials (or *Schrödinger factors*) such that $\Psi(x_t, t) \hat{\Psi}(x_t, t) = q_t(x_t)$; all other notation is defined in Section 2.1 of our paper.
> >
> > If we set $\Psi(x_t, t) = 1$ and $\hat{\Psi}(x_t, t) = q_t(x_t)$, then the necessary conditions on the potentials are met, and the backward SDE matches equation 2 in our paper, from which the probability flow ODE in equation 3 and the SD flow in equation 4 directly follow.
> >
> > ***
> >
> > Although there are other points from Reviewer SSdk that we will respond to in due course, we hope that we have resolved a number of the reviewer's concerns already.  We would welcome any additional comments from the reviewer in the meantime in response to our remarks thus far.
> >
> > ***
> >
> > [1] Chen, T., et al. "Likelihood training of Schrödinger bridge using forward-backward SDEs theory." arXiv preprint arXiv:2110.11291 (2021)
> >
> > [2] Su, Xuan, et al. "Dual diffusion implicit bridges for image-to-image translation." International Conference on Learning Representations. 2022.
> >
> > [3] Liu, Guan-Horng, et al. "I$^ 2$ SB: Image-to-Image Schrödinger Bridge." arXiv preprint arXiv:2302.05872 (2023).

---

> ### Author Response · Authors · 2023-05-09
> **Supplemental Response to Reviewer SSdk**
>
> We once again thank the reviewer for reading and responding to our work.  We hope that our previous comments addressed some of the reviewers concerns and that the reviewer finds the forthcoming substantially revised manuscript to be satisfactory.  A detailed list of the major changes will be posted along with the revision.
>
> In the meantime, we wish to address a few remaining points and preview some of the changes in the revision that we believe will be of interest to the reviewer.
>
> * The text has been reordered for a better flow.  A number of sections have been rewritten to better emphasize the contributions of the paper and call out key information.  We have also taken the reviewer's suggestion to move the discussion of general GAN dynamics to the appendix, where it works much better in the surrounding context of the Liouville equation and the related McKean-Vlasov process.
>
> * Regarding probability flow, our derivation of SD flow directly follows from this work.  However, we have expanded this discussion in Section 2.1 of the revised text.
>
> * The suggestion to take a closer look at SVGD and its generalizations is a good one.  In new experiments, we directly compare SD flow, MMD gradient flow, and SVGD under various experimental conditions and show that SD flow is the only algorithm to converge in all conditions.  We also include pointers to the literature connecting MMD gradient flow and SVGD.  These experiments have been placed in the main paper.  We still leave high-dimensional experiments on images to follow-up work, but we provide additional discussion to motivate that work.
>
> * We have included a new section discussing SD flow in the context of the Schrödinger bridge problem and show that it provides a solution to this problem.
>
> * We have rewritten or cut lines that were originally somewhat ambiguous in their wording, which we hope makes it clearer exactly what we intend to claim in this work.
>
> We hope that this resolves the reviewer's major concerns.  We welcome any additional comments or questions.

---

### Review · Reviewer_tqcW · 2023-04-28

**Summary Of Contributions:**

The paper deals with the so-called Score-Difference flow (SD flow) - the gradient flow with respect to KL divergence functional. The authors analyze SD flow from the theoretical point of view, establish the connection between SD flow and non-saturating GAN optimization as well as Denoising Diffusion models, and provide practical algorithms for SD flow modeling, which are based on substitution of original data distributions with noised versions of ones. The algorithms are validated on synthetic low and intermediate dimensions experiments.


**Audience:**

Yes

**Claims And Evidence:**

Yes

**Requested Changes:**

The general comments about what should be changed in the paper are in the Weaknesses section. Specific comments are below:

Page 3: The Stein’s identity should be properly introduced or cited.

Page 4: It was difficult for me to understand the last paragraph. What does it mean, that “prescribed dynamics for the clean data are the same as for the corrupted data”? The corrupted distributions are not the same as original distributions, so the dynamics should be slightly different. So, this paragraph should be reformulated more clearly.

Page 7: In the unnumbered equation, which follows equation (31), expectation sign is missing.

Page 8: I don’t understand the alternative GAN loss $\mathcal{L}^{\text{alt}}$, introduced in equation (32). Why does the optimal discriminator for the case of   $\mathcal{L}^{\text{alt}}$ induce the SD flow, as in the case of non-saturating loss?


**Strengths And Weaknesses:**

**Strengths**

I found the paper easy to follow. And, from my point of view, it is really important contribution of the work under consideration. In a rather simple and expressive narrative, the paper explains a rather difficult yet interesting topic. The Score-Difference flow that the authors propose is actually the gradient flow with respect to KL divergence functional. The existing literature devoted to this subject is either too theoretical or too specific and, more importantly, does not reveal the importance of gradient flows in machine learning community. Actually, the authors fill in the gap and show that SD flows secretly emerge in several generative modeling methods.
Concerning practical and technical achievements, the authors propose a novel approach to approximate the score difference, which prescribes to corrupt the original distributions under consideration with Gaussian noise and model the SD flow with respect to these corrupted distributions. Thereby, it becomes possible to estimate the score difference by samples, and, at the same time, approximately simulate the original score difference dynamics.

**Weaknesses**

In spite of a good overall impression on the paper, I found both theoretical and practical contributions to be limited. As I already note, the SD flows, which are the primal subjects of the paper, are known in literature and known to be the gradient flows with respect to KL divergence functional. There are existing papers which propose to utilize the gradient flows (w.r.t. KL or even f-divergence) for generative modeling [1], [2], [3], [4]. Of the special attention is paper [1]. The authors of [1] propose a framework for modeling gradient flows with respect to f-divergence, which reduces to KL divergence for $f(x) = x \log (x)$. Taking it into account, the results of Section 2.2 (from the paper under review) are in concordance with Theorem 2.1 from [1]. Besides, the optimal non-saturating GAN discriminator analysis (section 4.3) resembles Lemma 2.3 from [1]. Based on this, I believe, that [1] and, probably, [2], [3], [4] should be properly cited in the paper under consideration and the contributions of the latter should be stated more accurately. The comparison with the aforementioned papers is also highly desirable, because they could model the same gradient flows as considered in the paper under consideration, but utilize other techniques for density ratio estimation. By the way, [1] and [2] also exploit the idea to regress the parametric generator with squared $l_2$ norm loss in order to align it with the subsequent steps of gradient flow (Algorithm 2 from the paper under consideration).

Apart from the shortcomings described above, I think that the organization of the paper is another point to concern. It seems strange that all practical and algorithmical sections are located in the Appendix. I believe that the content of the paper should be organized in such a way that both theory and practice are somehow covered in the main part of the manuscript.

Lastly, I suppose that the practical validation of the approach is somehow limited. There are only proof-of-concept synthetic experiments in relatively small dimensions (additionally, the only “high”-dimensional experiment considered in the paper is to fit the Gaussian distribution, which doesn’t seem to be convincing). From my point of view, the authors should somehow persuade the readers that their approach indeed worth to be applied in real applications. It doesn't have to be high-dimensional image applications (I understand that the proposed method seems to scale poorly to high dimensions), but still, in my opinion, the machine learning is more about applications, not theory.

[1] Gao et. al., Deep Generative Learning via Variational Gradient Flow, http://proceedings.mlr.press/v97/gao19b/gao19b.pdf,  ICML’2019

[2] Gao et. al., Deep Generative Learning via Euler Particle Transport, https://arxiv.org/pdf/2110.02787.pdf , MSML’2021

[3] Feng et.al., Relative Entropy Gradient Sampler for Unnormalized Distributions, https://arxiv.org/pdf/2303.03714.pdf , arxiv’2021

[4] Heng et. al., Generative Modeling with Flow-Guided Density Ratio Learning, https://arxiv.org/pdf/2303.03714.pdf, arxiv’2023

---

> ### Author Response · Authors · 2023-04-29
> **Response to Reviewer tqcW**
>
> We thank the reviewer for this positive and helpful review.  We especially appreciate the reviewer's recognition of our contribution to the literature and our efforts to present what is ultimately theoretically challenging material in as accessible a manner as possible.  We are also grateful to the reviewer for excellent pointers to relevant work that we missed in our initial survey of the literature.
>
> We have already made a number of substantial changes to our working draft based on the reviewer's remarks and suggestions.  (We will post our revised manuscript once all reviews are in.)  A brief summary follows:
> * We have followed the reviewer's recommendation to reconsider the organization of the paper and have moved the algorithms and most experimental results to the main paper.  We are also adding additional comparisons to Stein variational gradient descent (SVGD), which, like MMD gradient flow, has a kernel-based specification with similarities to one of our implementations of SD flow.  This also better allows us to place the experiments we report in the context of this other work while also setting up follow-up work on higher-dimensional data using the alternative specifications of SD flow (e.g. denoising- and discriminator-based) defined in the paper.  We have also rewritten the introductions to the algorithm and experiment sections to better establish this context and, we hope, emphasize our theoretical and practical contributions.
> * We are sorry to have missed the work the reviewer referenced during our survey of the literature.  We found these references valuable and interesting and have included citations and discussion of their key results, including the leveraging of the $\ell_2$ loss in updating parametric generators on flow-perturbed targets, in several sections of the revised manuscript (e.g. Sections 2.2 and 4.1).  Indeed, we find the additional appearance of SD flow in the theoretical context of that work quite exciting, given that our derivation originally came from the study of the Langevin SDE and probability flow ODE.
> * We have included the relevant citations for Stein's identity and its multivariate generalization in Section 2.2.
> * We have rewritten the paragraph about clean- versus corrupted-data dynamics in Section 3.1.  Specifically, we point out that since $\mathbf{y} = \mathbf{z} - \epsilon$, then $\frac{d \mathbf{y}}{d \mathbf{z}} = \mathbf{I}$.  As a result, $\frac{d \mathbf{y}}{d t} = \frac{d \mathbf{y}}{d \mathbf{z}} \frac{d \mathbf{z}}{d t} = \frac{d \mathbf{z}}{d t}$.  Intuitively, the rationale here is essentially the same as the *reparameterization trick* that allows the propagation of a gradient through a stochastic node.
> * We thank the reviewer for catching the missing expectation operator above equation 31.  We have corrected this error.
> * We have added an additional line to make explicit the connection between the alternative GAN loss $\mathcal{L}^\text{alt}$ and SD flow when the discriminator is optimal.  Specifically, we show that the negative gradient of this loss produces the expected score difference, which induces the SD flow on individual points.
>
> I addition, we have added a new section (2.3), in which we show that SD flow provides a solution to the *Schrödinger bridge problem*.
>
> We thank the reviewer again for the helpful comments and suggestions and look forward to the reviewer's consideration of our revised manuscript once it is posted.

---

### Review · Reviewer_XeUD · 2023-05-08

**Summary Of Contributions:**



The authors introduce a new approach for implicit generative modeling (or for transforming/bridging samples from a source to a target distribution), called the **score difference flow**. The proposed framework transformation employs deterministic dynamics involving the difference of the logarithmic gradients of the two distributions.

The authors demonstrate that the proposed particle formulation reduces the KL divergence between target and transient density.
To ensure that the proposed score difference is computable they employ proxy distributions resulting from smoothing with multidimensional Gaussians  [9], and define sample dynamics for the noise corrupted instances of the involved distributions.

 The authors propose an alternative formulation for the score difference flow in terms of optimal denoising models of Gaussian corrupted distributions. A substantial part of the paper (and probably the most valuable contribution) is devoted to provide theoretical connections to generative adversarial networks (GANs) [Section 4], denoising diffusion models (DDMs) [Section 5], maximum mean discrepancy gradient flow (MMD) [Appendix D], which by itself is a valuable contribution.

The authors demonstrate the effectiveness of their proposed framework in the appendix through numerical experiments on toy models, which is sufficient given the other theoretical contributions of the paper.

Overall, the paper is technically sound, well-written, and provides valuable theoretical connections that are of interest to theTMLR audience.





**Audience:**

Yes

**Broader Impact Concerns:**

There is no Broader Impact Statement in the paper.
The proposed framework inherits the concerns on ethical implications of the works of diffusion-based generative modelling when applied to high dimensional image data.

**Claims And Evidence:**

Yes

**Requested Changes:**

**Suggestions for updates:**

**A]** I think the paper would become more complete if the authors make the connection of the score difference with the Langevin dynamics. In [1,5] the gradient flow structure of the Fokker-Planck was introduced with the KL divergence acting as a cost functional under the Wasserstein metric.

Thus for a (_time homogeneous_) diffusion process
$$ dX_{t} = - \\nabla U(X_{t}) dt + \\sqrt{2} dW_{t},$$
where $U(x) = - \\log p_{ss}(x)$ denotes a 'potential', and $p_{ss}$ is the target (stationary) density,
the score difference between a target (stationary) distribution and a instantaneous distribution
appears by rewriting the Fokker-Planck equation that describes the evolution of Langevin dynamics in terms of the difference of logarithmic gradients of the instantaneous and stationary probability densities


$$\partial q_t = \\nabla \\cdot \\left( q_t \\nabla U + \\nabla q_t \right) = -\\nabla \\cdot \\Big(  q_t \\left( \\nabla \\log p_{ss} -\\nabla \\log q_t  \\right)\\Big). $$


This can be seen as a Liouville equation with time-dependent velocity field the score difference $\\left( \\nabla \\log p_{ss} -\\nabla \\log q_t  \\right)$. The above equation corresponds to particle dynamics $dx = \\left( \\nabla \\log p_{ss} -\\nabla \\log q_t  \\right) dt$ in a similar vein as in the particle ODE framework (probability flow ODE) of [3, 2]. A recent related approach that claims to scale favorably with system dimension [10] considers a projected kernel density estimation for the score difference.


Here the authors consider a _time-inhomogeneous dynamics_ ($\mu(x,t)$, $\sigma(t)$) where in principle there might be no stationary probability density associated with the SDE, and thus one may not be able to define the sample trajectories using the gradient flow structure of the Fokker-Planck equation.


However I consider a mention of the equivalence of the two approaches in this time homogeneous case (when the target density is well captured by a potential) will provide a more clear picture of the contributions of the paper and the connections to the existing literature.


**B]** For the aforementioned reasons, as I understand, I would assume that the convergence rates of the score difference flow should be the same to those of the probability flow ODE/Fokker-Planck approach. However in the discussion the authors mention *"If our goal is to get from point A to point B as quickly as possible, .."*. I would consider that alternative transport based approaches may show faster convergence properties to the target distribution. For example the very recent framework of [12] claims to provide possibly even one step interpolation between the two densities. To be on the safe side, I would reformulate this statement.


**C]** Here the authors consider the score difference between a stationary distribution (p) and a transient distribution (q) for the definition that reduces the KL divergence between the two probability densities. This resembles the optimal control formulation for diffusion processes introduced in [6, 7], where the authors consider the score difference between *transient* probability densities in an optimal control setting [8] that optimally minimises the KL divergence between the two transient solutions. It might be interesting to discuss the connections of the two frameworks.


**D]** In the last paragraph the authors mention *"SD flow supplies a link between IGM methods that..[provide] high sample quality, mode coverage, and fast sampling"*. However I am not entirely convinced about this statement. I think the statement should be either reformulated, or the authors should provide links to numerical experiments/links that validate their claims.

**E]** I wonder whether some variant of the score difference flow that employs some state-of-the-art score matching frameworks may be more suitable for image generation. I am not asking for additional experiments on generating images (or other high dimensional data), however I think some statement in conjunction to the appendix E.4 would be valuable to assess the usefulness of the proposed method for practical applications.

**Minor suggestions:**

- The flow of the text would become more coherent if the Section 5 (Relation to Denoising Diffusion Models) follows Section 3, before starting the extensive section on the relation to GANs.

- I wonder whether it would be useful to extend the length of the article and include the numerical experiments in the main text, and also bring the relation to the MMD flow forward. Then the relations to GANs, DDMs, and MMD will be clustered.

- The deterministic ODE process for sampling transient densities associated with SDEs was first introduced in [3], and was then applied in [2] and named “probability flow ODE”. There are already hints of this expression already in [4].

- The figures would improve if the text in axis labels/ticks had bigger fonts.


**References:**

[1] Jordan, R., Kinderlehrer, D., \& Otto, F. (1998). The variational formulation of the Fokker--Planck equation. *SIAM journal on mathematical analysis*, *29*(1), 1-17.

[2] Song, Y., Sohl-Dickstein, J., Kingma, D. P., Kumar, A., Ermon, S., \&
 Poole, B. (2020). Score-based generative modeling through stochastic
differential equations. *arXiv preprint arXiv:2011.13456*.

[3] Maoutsa, D., Reich, S., \& Opper, M. (2020). Interacting particle
solutions of Fokker–Planck equations through gradient–log–density
estimation. *Entropy*, *22*(8), 802.

[4] Otto, F., \& Villani, C. (2000). Generalization of an inequality by
Talagrand and links with the logarithmic Sobolev inequality. *Journal of Functional Analysis*, *173*(2), 361-400.

[5] Otto, F. (2001). The geometry of dissipative evolution equations: the porous medium equation.

[6] Maoutsa, D., \& Opper, M. (2021). Deterministic particle flows for constraining SDEs. *Machine Learning and the Physical Sciences, Workshop at the 35th Conference on Neural Information Processing Systems (NeurIPS).*

[7] Maoutsa, D., \& Opper, M. (2022). Deterministic particle flows for constraining stochastic nonlinear systems. *Physical Review Research*, *4*(4), 043035.

[8] Kappen, H. J., Gómez, V., \& Opper, M. (2012). Optimal control as a graphical model inference problem. *Machine learning*, *87*, 159-182.


[9] Zhang, M., Hayes, P., Bird, T., Habib, R., \& Barber, D. (2020, November). Spread divergence. In International Conference on Machine Learning (pp. 11106-11116). PMLR.

[10] Wang, Y., Chen, P., \& Li, W. (2022). Projected Wasserstein gradient descent for high-dimensional Bayesian inference. SIAM/ASA Journal on Uncertainty Quantification, 10(4), 1513-1532.

[11] El Moselhy, T. A., & Marzouk, Y. M. (2012). Bayesian inference with optimal maps. *Journal of Computational Physics*, 231(23), 7815-7850.

[12] Liu, X., Gong, C., & Liu, Q. (2022). Flow straight and fast: Learning to generate and transfer data with rectified flow. *ICLR2023.*

**Strengths And Weaknesses:**

**Strengths:**

- Well written manuscript provides a good overview of the recent contributions for sampling intractable distributions and connects the proposed framework with existing literature.

- The advantage of this approach compared to usual Langevin dynamics that admit a sampling with probability flow ODEs is that the target distribution in this setting does not need to be analytically defined (e.g. in terms of a potential), but can rather be represented by samples as in the numerical experiments in the Appendix.

**Weaknesses:**

- Although the authors connect the score difference flow to diffusion models, they do not provide the straightforward connection to the gradient flow dynamics of Fokker-Planck equation.

- There may be practical/numerical limitations in the practical uses of the framework for high-dimensional data generation, but this by itself is not a problem. However it may be useful for the authors to mention which parts of the proposed framework hinder its application to high-dimensional data (if there is indeed any such limitation).

---

> ### Author Response · Authors · 2023-05-09
> **Response to Reviewer XeUD**
>
> We thank the reviewer for this encouraging and insightful review and the helpful suggestions contained therein.  We will upload a substantially revised draft shortly, with pointers to the main changes.  In the meantime, we will preview these changes below in response to the reviewer's comments.
>
> **A.** A discussion of the connection to the Fokker-Planck equation was originally cut from an earlier draft of Section 2.1 in order to arrive at the score difference derivation more quickly.  However, we agree that this connection is important, so we have restored this discussion and added a reference to [1] and a mention of its main result.
>
> **B.** Our numerical experiments suggest that SD flow does indeed converge more quickly than competing methods (most notably MMD gradient flow and Stein variational gradient descent), but we have not performed a formal convergence analysis.  For this reason, we have taken the reviewer's suggestion and cut the line discussing getting from A to B "as fast as possible."
>
> **C.** We thank the reviewer for this reference to interesting work on KL control problems.  We have included a reference to [2] at the end of Section 2.2 in the revised paper.
>
> **D.** We regret that the original wording of this statement was somewhat ambiguous.  We have rewritten this part to read:
> >SD flow supplies a link between IGM methods—namely GANs and diffusion models—that collectively perform well on all three desiderata of the so-called *generative learning trilemma*: high sample quality, mode coverage, and fast sampling.  Inserting SD flow as an alternative, non-adversarial target-generation module within generator training, replacing the need for an optimal discriminator, could lead to the development of "triple threat" models that produce high-quality, diverse output in a single inference step.  We look forward to developments in this direction.
>
> Here we intended only to say that the connection among SD flow, diffusion models, and GANs hints at a unified approach that can address the trilemma, which we hope our work to provide a step toward.
>
> **E.** This is a good point.  We have added a paragraph discussing this at the end of Appendix C.4 (which was E.4 in the submitted draft).
>
> Regarding the minor suggestions:
> * Reordering the text is an excellent suggestion.  The connection to diffusion models now directly follows Section 3.2 and comes before the section on GANs.
>
> * Our original choice to place the experiments in the appendix was made to emphasize the theoretical component of the paper, but we agree that the work is better served by including experiments in the main paper.  (We think it also makes the ending of the main text feel less abrupt.) Section 7 in the revised draft is devoted to reporting new experiments, namely head-to-head comparisons of SD flow, MMD gradient flow, and SVGD under various experimental conditions.  We show that SD flow is the only algorithm to converge under *all* tested conditions.  Additional experiments are reported in Appendix C.
>
> * Thank you for reminding us of this reference.  We have included a reference to [3] in Section 2.1.
>
> * We generated new figures for the experiments reported in Section 7.  We hope that the labels are more legible, but we would be happy to make further adjustments if necessary.
>
> We have also inserted new material (Section 2.3) showing that SD flow provides a solution to the *Schrödinger bridge* problem.  The main paper also now closes with a broader impact statement.
>
>
>
> [1] Jordan, R., Kinderlehrer, D., & Otto, F. (1998). The variational formulation of the Fokker--Planck equation. SIAM journal on mathematical analysis, 29(1), 1-17.
>
>
> [2] Maoutsa, D., & Opper, M. (2022). Deterministic particle flows for constraining stochastic nonlinear systems. Physical Review Research, 4(4), 043035.
>
> [3] Maoutsa, D., Reich, S., & Opper, M. (2020). Interacting particle solutions of Fokker–Planck equations through gradient–log–density estimation. Entropy, 22(8), 802.

---

### Decision · Action_Editors · 2023-06-23

**Recommendation:** Accept with minor revision

**Comment:**

Overall I believe this work satisfies the TMLR acceptance criteria and thus should be accepted for publication at TMLR.  While I share the frustration of some of the reviewers (especially Reviewer SSdk) that the paper could be noticeably improved, in particular by providing more serious empirical evaluations, I do not think this undermines its adherence to the TMLR criteria in its current form.  The substantial updates that the author's made in the revision have corrected most of the other issues that I think would have prevented acceptance.

I would though like to make the following requests for minor revision before the final acceptance of the work:
- Provide some discussion on the current limitations of the approach for practical use in its current form and what would be needed to use it for real problems.  I am worried at the moment that it might be difficult for others to follow up from a more practical standpoint, and I think the current algorithmic limitations need to be made clear so that it does not end up being a "blocker" to future work (i.e. if the current paper is being accepted on the basis of being a theoretical contribution without providing proper empirical evaluation, it should not get in the way of future work being published that uses similar ideas from a more practical standpoint).
- Addition of the following citation requested by Review XeUD (who is not an author of the work): Wang, Y., Chen, P., & Li, W. (2022). Projected Wasserstein gradient descent for high-dimensional Bayesian inference. SIAM/ASA Journal on Uncertainty Quantification, 10(4), 1513-1532.
- If possible, change the colour scheme of the plots to avoid red-green clashes for people
- Try to address the following comments made by Reviewer tqcW in their official recommendation:
> I am not fully satisfied with the clarifications given by authors to some of my questions. In particular, the phrase, that “the prescribed dynamics for the clean data y under SD flow are the same as for the corrupted data” (last paragraph of the Section 3.1) sounds slightly strange for me, because the dynamics for noised data is not the same (but close to) the dynamics of the original data. And the reparametrization trick analogy seems not applicable in this case. Secondly, I still don’t understand the idea with alternative GAN loss
$\mathcal{L}^{alt}$ (eq. (37)). Why does the optimal discriminator for this particular GAN loss is the log density ratio?



**Audience:**

All reviewers believed that the audience criterion was satisfied and I agree with this view.

Though I think the inclusion of more serious empirical evaluations would substantially improve its potential significance and impact (and its omission would likely lead me to a reject recommendation for one of the top ML conferences), there is certainly still a contribution to the literature without this and there will be people in the TMLR community interested in these.

**Claims And Evidence:**

I believe that the claims are overall sufficiently supported to satisfy this criterion.

No reviewers had any significant complaints about the correctness of theoretical work or its relationship to the high-level claims made.  All the reviewers (and myself) agree that the empirical evaluation is rather lacking, but there were differences in opinion on whether the paper should be rejected based on this or not (with two reviewers arguing for acceptance and one rejection).  My view is that, while this is certainly a significant weakness of the work, the paper (at least in its updated version) does not actually make any notable empirical claims about the approach and is pitched predominantly as theoretical work.  As such, I do not believe that there are currently unsupported claims, or that the paper is misleading in its contributions.  Thus I believe it satisfies the requirements for TMLR on this aspect.

---

> ### Author Response · Authors · 2023-07-18
> **Response to Recommendation**
>
> We thank the reviewers and action editor for the consideration that led to this decision.  We are pleased that the majority opinion was that the theoretical contributions of this work stand on their own.  At the same time, we acknowledge the value that a more comprehensive exploration of the experimental aspects of SD flow would bring, which we look forward to as the subject of follow-up work.
>
> In addition to a number of minor changes introduced in a top-to-bottom edit, the following changes have been made to the camera-ready version of the text in response to the AE's requests:
> * Section 3.2 has been retitled "Limitations and Alternative Formulations of SD Flow" and now features a longer discussion of the potential limitations of the kernel-based definition of SD flow while providing commentary on applying the denoiser-based definition on high-dimensional data.  We believe that this rewritten section will make the application of SD flow in high dimensions clearer to interested readers.
> * A reference to and brief discussion of projected Wasserstein gradient descent (Wang, Chen & Li, 2022) has been added to Section 3.2.
> * The color scheme of all color figures has been changed from red-green to red-blue.  Thank you for this suggestion.
> * Reviewer tqcW was justified in being skeptical of our initial responses to these questions, since in both cases, the original answers were either incomplete or otherwise needed adjustment.  We provide a different, more detailed discussion of why we set $\Delta y = \Delta z$ at the end of Section 3.1, which is also more explicit about the assumptions involved and the resulting interpretation of $\Delta y$.  And in Section 5.2, we much more carefully derive the results showing that the alternative loss induces the SD flow when the discriminator is optimal, which also corrects a minor error in the definition of this loss.
>
> Once again, we thank the reviewers and AE for their time and consideration.